METHODS

# MIAAIM: Multi-omics image integration with dimensional reduction for tissue state mapping

Joshua M. Hess[1,2], Richard K. Dzeng[1,3], Iulian Ilieş[4], Denis Schapiro[5,6,7,8,9], John J. Iskra[1,10], Divya Mirgh[1], John Nam[1], Erin H. Seeley[11], David E. Verrill[12], Walid M. Abdelmoula[13¤], Michael S. Regan[13], Georgios Theocharidis[14], Chin Lee Wu[15], Aristidis Veves[14], Nathalie Y. R. Agar[13,16], Ann E. Sluder[1], Mark C. Poznansky[1*ठ], Ruxandra F. Sîrbulescu[1,17*ठ], Patrick M. Reeves[1,18*ठ]

1 Vaccine and Immunotherapy Center, Massachusetts General Hospital, Harvard Medical School, Boston, Massachusetts, United States of America, 2 Tri-Institutional Training Program in Computational Biology and Medicine, Weill Cornell Medicine, New York, New York, United States of America, 3 Department of Computer Science, Tufts University, Medford, Massachusetts, United States of America, 4 Healthcare Systems Engineering Institute, Northeastern University, Boston, Massachusetts, United States of America, 5 Laboratory of Systems Pharmacology, Harvard Medical School, Boston, Massachusetts, United States of America, 6 Broad Institute of Harvard Medical School and Massachusetts Institute of Technology, Boston, Massachusetts, United States of America, 7 Faculty of Medicine, Institute for Computational Biomedicine, Heidelberg University Hospital and Heidelberg University, Heidelberg, Germany, 8 Institute of Pathology, Heidelberg University Hospital, Heidelberg, Germany, 9 Translational Spatial Profiling Center, Heidelberg, Germany, 10 Department of Mathematics, Emory & Henry University, Emory, Virginia, United States of America, 11 Department of Chemistry, University of Texas, Austin, Texas, United States of America, 12 Chemistry and Chemical Biology, Barnett Institute for Chemical & Biological Analysis, Northeastern University, Boston, Massachusetts, United States of America, 13 Department of Neurosurgery, Brigham and Women's Hospital, Harvard Medical School, Boston, Massachusetts, United States of America, 14 The Rongxiang Xu Center for Regenerative Therapeutics, Beth Israel Deaconess Medical Center, Harvard Medical School, Boston, Massachusetts, United States of America, 15 Department of Pathology, Massachusetts General Hospital, Harvard Medical School, Boston, Massachusetts, United States of America, 16 Department of Radiology, Brigham and Women's Hospital, Harvard Medical School, Boston, Massachusetts, United States of America, 17 Department of Neurology, Massachusetts General Hospital, Harvard Medical School, Boston, Massachusetts, United States of America, 18 Department of Surgery, Massachusetts General Hospital, Harvard Medical School, Boston, Massachusetts, United States of America

ठ MCP, RFS, and PMR contributed equally to and are co-corresponding authors for this manuscript
¤ Current address: Invicro, LLC, Boston, Massachusetts, United States of America
* mpoznansky@mgh.harvard.edu (MCP); rsirbulescu@mgh.harvard.edu (RFS); patrick_reeves@hms.harvard.edu (PMR)

## Abstract

High-parameter tissue imaging enables detailed molecular analysis of single cells within their spatial environment. A current challenge to more complete tissue and single-cell spatial profiling is *in situ* data alignment across imaging platforms that quantify multiple types of biomolecules at differing resolutions. Here, we describe MIAAIM (Multi-omics Image Alignment and Analysis by Information Manifolds), a modular framework to align and process data from separate imaging technologies with distinct imaging resolutions and data complexity. MIAAIM is designed to be applied to align and analyze images of clinical biopsies from histological staining, imaging mass cytometry, and mass spectrometry imaging. A key advantage of the MIAAIM approach is its capacity to identify unbiased molecular phenotypes that

**Data availability statement:** All newly generated data from this study are available at https://dataverse.harvard.edu/dataset.xhtml?persistentId=doi:10.7910/DVN/YU6V7E and https://dataverse.harvard.edu/dataset.xhtml?persistentId=doi:10.7910/DVN/PUWHPQ. Code availability statement: All software and code that produced the findings of the study, including all main and supporting figures, are publicly available via GitHub at https://github.com/MGH-VIC/miaaim-python-docker. A user manual and training guide to get started using the MIAAIM software and associated workflows is available at https://mgh-vic.github.io/. The code base is available at https://github.com/MGH-VIC/miaaim-python-docker.

**Funding:** The study was supported by Vaccine and Immunotherapy Center Innovation and Education Funds at Massachusetts General Hospital, including salary to J.M.H, R.K.D., D.M., A.E.S., M.C.P., P.M.R. and R.F.S. The study was also supported by a Congressionally Directed Medical Research Program award (W81XWH-20-1-0301), including salary to J.M.H and P.M.R. J.M.H. received salary from a National Science Foundation Graduate Research Fellowship under grant 1746886. D.S. received salary support by the University of Zurich BioEntrepreneur-Fellowship (BIOEF-17-001), a Swiss National Science Foundation Early Postdoc Mobility fellowship (P2ZHP3_181475, BMBF (01ZZ2004) and a Damon Runyon Fellow supported by the Damon Runyon Cancer Research Foundation (DRQ-03-20). N.Y.R.A received partial salary support from NIH grants U54-CA210180 and P41-EB028741. The funders had no role in study design, data collection and analysis, decision to publish, or preparation of the manuscript.

**Competing interests:** I have read the journal's policy and the authors of this manuscript have the following competing interests: D.S. is a scientific consultant to Roche Glycart AG; none of these relationships are directly related to the topic of this study. R.F.S., P.M.R. and M.C.P. are scientific founders of PDTx and consultants to Ovation.io. R.F.S., J.M.H., P.M.R, and M.C.P. are authors on patent PCT/US21/48928 applied for by the General Hospital Corporation covering the described methods.

correlate with cell identities and states determined using high-resolution targeted immunodetection. In a large diabetic foot ulcer (DFU) biopsy, this strategy allowed the identification of unique molecular characteristics of infiltrating immune cells as a function of local tissue health. In multi-core tissue microarrays (TMAs) of prostate cancer, MIAAIM allowed the classification of adjacent tumor grades with high accuracy, with over 90% of classification signal sourced from spatial features, generated from segmented cells across multiple imaging modalities while revealing novel cell/immune signatures of the disease state. MIAAIM provides a disease and cell type agnostic general framework to construct multimodal tissue imaging datasets, yielding novel insights into the association of molecular analytes with cell subsets and their activation states for the analysis of complex tissue states.

## Author summary

Integrating tissue imaging across different modalities and resolutions is a significant challenge that profiles the single cell and its local microenvironment within the tissue architecture. Here, we describe MIAAIM (Multi-omics Image Alignment and Analysis by Information Manifolds), a modular framework to align and combine data across imaging resolutions and multimodal imaging stacks. MIAAIM was used to integrate optical, imaging mass cytometry (IMC), and mass spectrometry imaging (MSI) of diabetic foot ulcer (DFU) clinical biopsies and prostate cancer tissue microarrays (TMAs). On the DFU sample, we explored immune niches within the tissue microenvironment. Using prostate cancer TMAs we generated single cell and cellular network profiles that led to state-of-the-art classification accuracies of Gleason score. MIAAIM provides a generalizable framework to integrate complex multimodal tissue imaging datasets, yielding novel insights into the association between molecular analytes and cellular populations across tissue structures.

## Introduction

High-parameter imaging technologies are increasingly available to capture aspects of complex tissue states. These technologies range from untargeted methods, such as mass spectrometry imaging (MSI) [1] and whole transcriptome spatial transcriptomics [2,3], which provide broad molecular profiles, to targeted, multiplexed approaches utilizing panels of transcript or antibody probes [4–10]. Label-free detection methods allow for hypothesis-free (agnostic) detection of biomolecules including lipids, metabolites, glycans, and protein-derived peptides [11]. Correspondingly, high-parameter probe-based methods can detail cellular populations and functional states within a tissue at cellular or sub-cellular resolution. High-parameter, high-resolution imaging enables multimodal interrogations of cell and tissue organization in a deep spatial context with potential to identify connections across molecular classes and biological scales. Recent advances enable profiling of multiple target classes *in situ* (e.g., [12]).

Multimodal data are more commonly acquired [13] using adjacent sections [13,14], however these approaches often differ in imaging resolution and can introduce tissue deformations. Consequently, image alignment strategies must consider cross-modality challenges. Co-registration of images often relies on manual alignment to a single modality (e.g., hematoxylin & eosin (H&E), immunofluorescence) or use of reference markers shared across modalities with some accommodations for tissue deformations, optical distortions, and other technical variations. More recent co-registration approaches incorporate computational and machine learning tools to enhance performance and enable alignment without reference [15–21]. However, an open-source, reproducible workflow to build multimodal imaging datasets and extract associated single-cell data that generalizes to a growing repertoire of modalities is currently not available.

Here, we describe MIAAIM (Multi-omics Image Alignment and Analysis by Information Manifolds), a modular framework with reproducible workflows to align data across high-parameter imaging technologies and convert aligned images into multimodal single-cell data. MIAAIM introduces a manifold-based image registration approach, designed to compress and integrate data with diverse densities and spatial resolutions, that can be generalized to multiple high-parameter bioimaging systems. Additional modules in MIAAIM perform necessary cell segmentation and quantification steps to convert aligned pixel-level imaging data to multimodal single-cell measurements.

We demonstrate MIAAIM using imaging data derived from a combination of mass spectrometry-based imaging technologies, including imaging mass cytometry (IMC) and mass spectrometry imaging (MSI), anchored to classical histological H&E staining in two separate contexts. First, biopsy sections of diabetic foot ulcer were imaged fully by MSI, with select regions of interest (ROI) imaged by IMC. Single-cell analyses were performed to determine alterations in cellular composition and cellular phenotypic states as tissue transitions from wound bed to healthy tissue. To subsequently demonstrate MIAAIM's robustness and applicability to clinically relevant tissues, we profiled 30 prostate cancer (PC) biopsies from a tissue microarray (TMA). The resulting multimodal spatial information was used to develop a model that classifies histopathological grade, revealing prognostic biological and immune signatures of disease. Together, these applications demonstrate the ability of MIAAIM to co-register multi-modal, high parameter images and its subsequent utility in identifying spatial features linked to biological and clinical observations.

## Results

### MIAAIM workflow

We developed MIAAIM as a multi-source image analysis software in Python comprised of 5 processing stages (**Fig 1**). The first two stages (i) high-dimensional image preprocessing and compression (HDIprep), and (ii) high-dimensional image registration (HDIreg) are applied sequentially to align multimodal images. The remaining stages generate probability maps as input for cell segmentation, perform segmentation, and extract corresponding multimodal single-cell data using aligned images output from the HDIreg module.

Image alignment in MIAAIM begins with two or more assembled images [22,23] or spatially resolved data sets (**Fig 1a**). Currently, the size and standardized format of assembled images vary by technology. For example, cyclic fluorescence-based methods (e.g., CODEX [6], CyCIF [8,9]) assemble BioFormats/OME-compatible [24] 20–60-plex full-tissue images after correcting uneven illumination (e.g., BaSiC [25]) and stitching tiles (e.g., ASHLAR [26]). Other imaging methods acquire 20–100-plex data within specified regions of interest (ROIs) (e.g., MIBI [4], IMC [5]) in the OME format. Additional methods quantify thousands of parameters at rasterized locations on full tissues or ROIs and are not stored in BioFormats/OME-compatible file formats (e.g., the imzML format [27] often used for MSI data). MIAAIM accepts a variety of input data formats, and its modular design enables it to incorporate additional formats tailored to new imaging modalities. Additional cross-validation of optimized methods was performed using data sets collected from imaging tissue sections from tonsil and prostate biopsies (S1, S2, and S3 Figs).

Regardless of technology, assembled images contain high numbers of heterogeneously distributed parameters, which precludes comprehensive and manually guided image alignment. In addition, high-dimensional imaging produces large

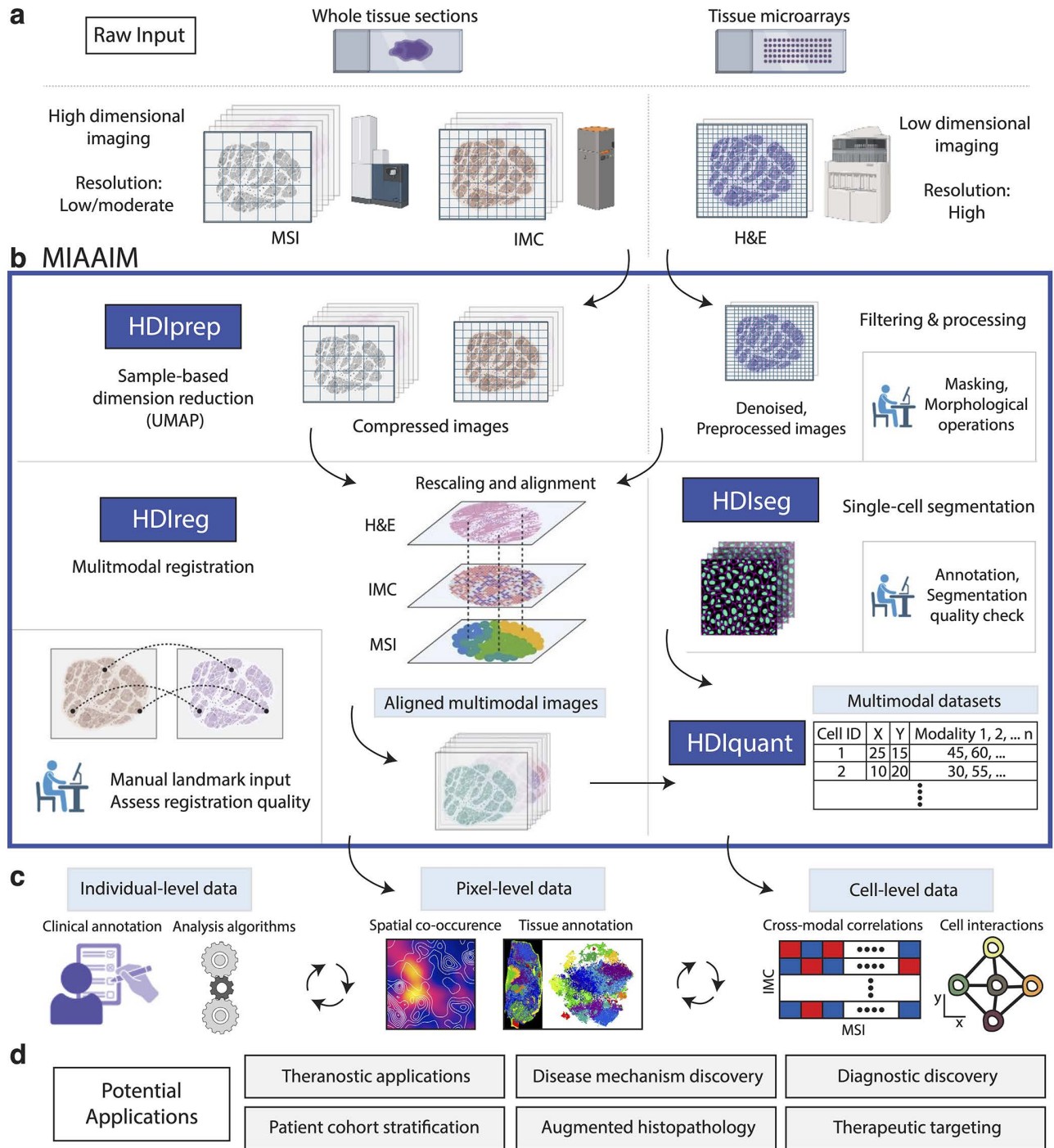

**Fig 1. Summary of MIAAIM workflow and key outputs. a.** Multidimensional datasets, such as mass spectrometry imaging (MSI) and imaging mass cytometry (IMC) together with classical histology H&E data from whole tissue sections or tissue microarrays (TMAs) serve as inputs to MIAAIM. **b.** The MIAAIM software includes algorithms and workflows for processing dataflows associated with high-parameter images (> 3 channels) through the HDIprep module for pixel-level image compression (left), and low-parameter images (≤ 3 channels) through image denoising and filtering (right). Supervised quality checks and user input are utilized at multiple points of the processing (human-in-the-loop). HDIReg is used to rescale and align pre-processed images using a manifold-based image registration approach. To bypass or augment automatic alignment methods, landmarks can also be set manually and used to guide the alignment process. Single-cell resolved modalities can be used to create cell segmentation masks in the HDIseg

module to extract multimodal single cell profiles in the HDIquant module. **c.** Pixel or cell-level multimodal data can then be processed by clustering, neighborhood interaction and related to external patient measures. **d**. Outcomes of this analysis can be used for a variety of applications, from monitoring response to therapeutic interventions to mechanistic and diagnostic discovery of novel targets. Created in BioRender. Sirbulescu, **R.** (2026) https://BioRender.com/yyq3k3t.

feature spaces that challenge unsupervised methods. The HDIprep workflow generates compressed images that preserve multiplex salient features to enable cross-technology alignment while reducing computational complexity (**Fig 1b**, HDIprep). For images acquired from histological staining, HDIprep implements parallelized smoothing and morphological operations that can be applied sequentially for preprocessing.

Image registration with HDIreg produces transformations to combine different imaging modalities within the same spatial domain (**Fig 1b**, HDIreg). HDIreg uses Elastix [28], a parallelized image registration library, to calculate between-image mappings. HDIreg is optimized to transform multichannel images with minimal memory use, in addition to supporting histological stains. Image resizing, padding, and trimming of borders prior to applying image transformations are also incorporated.

Aligned data output from HDIreg can be analyzed at the pixel level. Alternatively, they are well-suited for established single-cell and spatial neighborhood analyses [29] after cell segmentation and quantification of multimodal single-cell measures, such as average protein expression. Single-cell segmentation in MIAAIM currently uses custom Python modules implementing Ilastik [30] in conjunction with CellProfiler [31]. Multimodal data quantification is performed using an adaptation of the MCMICRO quantification module on aligned pixel-level images (interpolated data for transformed images) [22,23].

MIAAIM's algorithms are nonparametric, rather than deep learning based, allowing for generalization to multiple imaging systems through technology-agnostic alignment on a case-by-case basis (S1 Table). However, this classical, optimization-based registration approach is an iterative process, requiring parameter-tuning. This approach creates a substantial challenge for reproducible image alignment and subsequent single-cell quantification. To remove language-specific dependencies and document human-in-the-loop processing steps in accordance with the FAIR (findable, accessible, interoperable, and reusable) [34] data stewardship principles MIAAIM was Docker [32] containerized with parameter logging functionality included, inspired by Nextflow [33] and MCMICRO [22].

The following sections describe algorithms unique to MIAAIM and their application to multimodal imaging data sets across technologies and diverse tissue specimens.

## High-dimensional image compression with HDIprep

To compress high-parameter images, HDIprep performs dimensionality reduction on pixels using Uniform Manifold Approximation and Projection (UMAP) [34] (**Fig 2a**). A stringent comparison of dimensionality reduction algorithms was performed using imaging data sets of human DFU, prostate cancer, and tonsil tissue biopsies acquired using MSI, IMC, and H&E stain. Based on dimensionality reduction benchmarks, UMAP consistently outperformed competing algorithms in its robustness to noise and ability to efficiently preserve data complexity while capturing morphological structure (S1, S2, and S3 Figs and S3 Note).

HDIprep retains data complexity with the fewest degrees of freedom necessary by estimating an "optimal" dimensionality for pixel embeddings. The information captured by UMAP pixel embeddings (S4 Note, **Definition 1,** cross-entropy) is computed across a range of embedding dimensionalities, and the first dimension where the observed cross-entropy approaches the asymptote of an exponential regression fit is selected. This fit is based on a heuristic power-law relationship between distances within neighborhoods of points in the embedding space to original, high-dimensional distances in an idealized scenario (S4 Note**).** However, with each embedding dimensionality, exact cross-entropy calculations scale

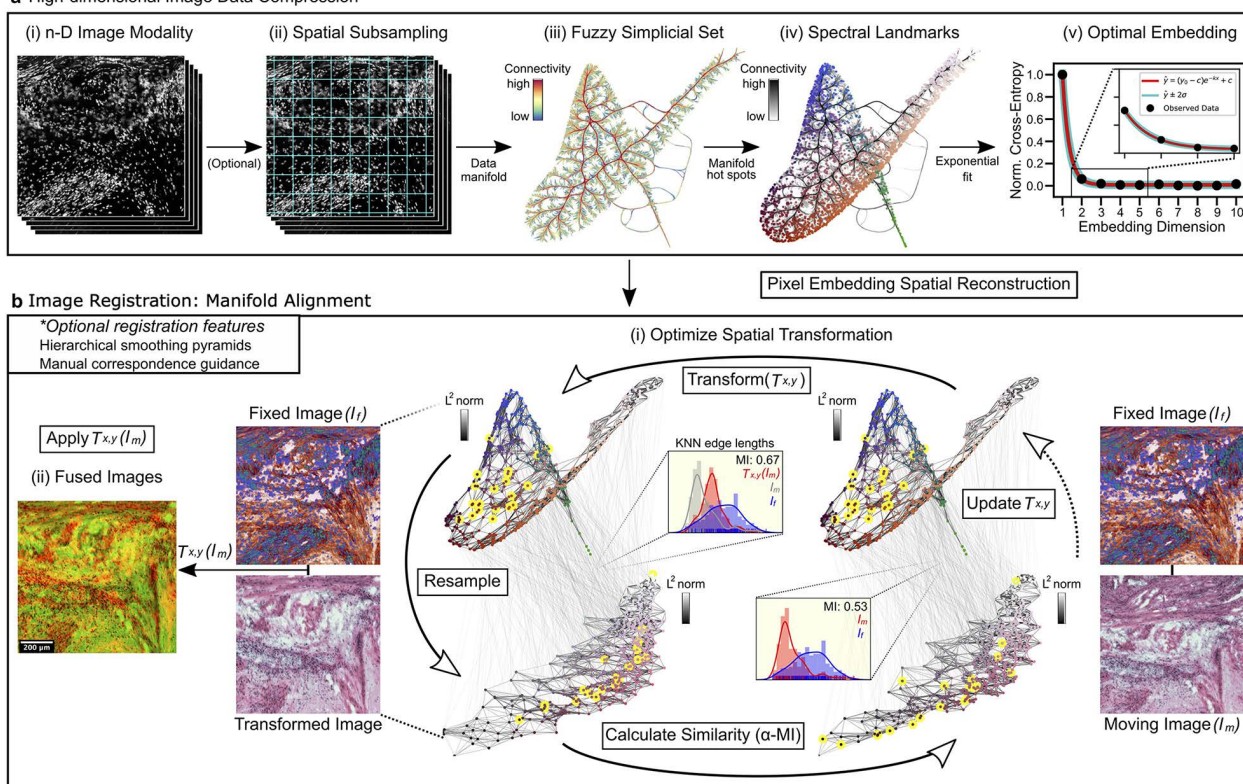

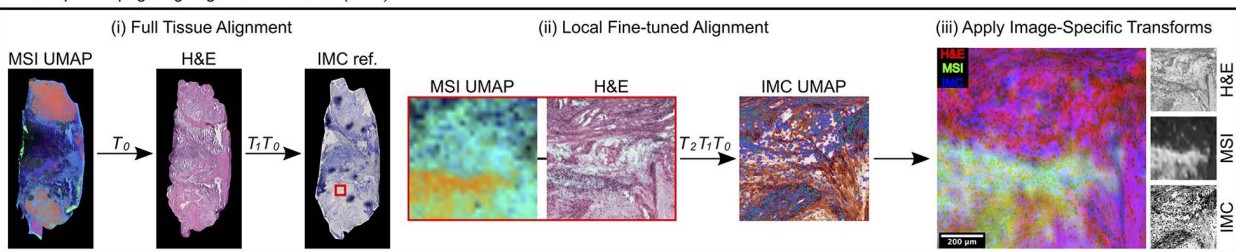

**Fig 2. HDIprep compression and spatially constrained manifold alignment with HDIreg. a.** HDIprep compression algorithm. **(i)** High-dimensional modality. **(ii)** Spatial subsampling. **(iii)** Data manifold represented as a fuzzy simplicial set (UMAP). Edge bundled connectivity of the manifold is shown on two axes of the estimated "optimal" embedding (*fractal-like structure may not reflect biologically relevant features). **(iv)** High-connectivity landmarks identified with spectral clustering. **(v)** Spectral landmarks are embedded into a range of dimensionalities and exponential regression identifies steady-state dimensionalities. Pixel locations are used to reconstruct compressed image. **b.** HDIreg manifold alignment. **(i)** Spatial transformation is optimized to align moving image to fixed image. KNN graph lengths between resampled points (yellow) are used to compute $\alpha$-MI. Edge-length distribution panels show Shannon MI between distributions of intra-graph edge lengths at resampled locations before and after alignment ($\alpha$-MI converges to Shannon MI as $\alpha \longrightarrow 1$). MI values show increase in information shared between images after alignment. KNN graph connections show correspondences across modalities. **(ii)** Optimized transformation aligns images. Shown are results of transformed H&E image (green) to IMC (red). **c.** Example alignment. **(i)** Full-tissue MSI-to-H&E registration produces $T_0$. **(ii)** H&E is transformed to IMC full-tissue reference, producing $T_1$. **(iii)** ROI coordinates extract underlying MSI and IMC data in IMC reference space. **(iv)** H&E ROI is transformed to correct in IMC domain, producing $T_2$. Final alignment applies modality-specific transformations. Shown are results for an IMC ROI and interpolated data for H&E and MSI modalities.

quadratically with the number of pixels. Therefore, HDIprep embeds landmarks in the pixel manifold representative of its global structure to reduce computational complexity (S4 Fig).

Pixel-level dimensionality reduction is computationally expensive for images with large numbers of pixels. To reduce compression time while preserving quality, we developed a subsampling approach to embed a spatially representative subset of pixels prior to spectral landmark selection and project out-of-sample pixels into embeddings (S5, S6, and S7 Figs). HDIprep combines all optimizations with a neural-network UMAP implementation [35] to scale to whole-tissue images. We demonstrate its efficacy on publicly available [36] 44-channel CyCIF images containing ~100 and ~256 million pixels (S8 Fig). Thus, HDIprep presents an objective, pixel-level compression method applicable to multiple modalities (**Methods, Algorithm 1**).

**High-dimensional image registration (HDIreg)**

MIAAIM connects the HDIprep and HDIreg workflows with a manifold-based image alignment approach. To align pixel manifolds, the methodology to compute the *intrinsic* $\alpha$-entropy [37,38] of data manifolds using entropic graphs is extended to estimated optimal UMAP embeddings from HDIprep and used for image registration through the entropic graph-based Rényi $\alpha$-mutual information ($\alpha$-MI) [39] (**Methods, HDIreg** and S4 Note). HDIreg searches for a transformation that maximizes image-to-image (manifold-to-manifold) $\alpha$-MI (**Fig 2b**).

Similarity measures to compare separate images, such as histogram-based mutual information or cross-correlation, are computationally efficient and often used for multimodal image alignment [40,41]. However, they require equal numbers of channels across modalities and prior knowledge of a correspondence between them, as they are not invariant to channel indexing (i.e., relabeling channels will produce different values). Similarly, while previous works have embedded multimodal 2-D data as 3-D pixel embeddings for image alignment [42,43], they first apply red/green/blue (RGB)-to-grayscale conversions before optimizing these pixel-intensity based similarity measures. These conversions constrain the information content retained in embeddings (by dimensionality) and impose further bias in registration, as grayscale conversions are not invariant to RGB channel labeling and as a result, embedding dimension labels can be assigned arbitrarily. These shortcomings also apply to the *extrinsic* Rényi α-mutual information, which has been used for image alignment, but requires equal numbers of channels across images, as it relies directly on pixel intensity-based histograms.

In contrast, the $\alpha$-MI generalizes to arbitrary dimensionalities by considering distributions of *k*-nearest neighbor (KNN) graph lengths of compressed (i.e., embedded) pixels, rather than comparing pixel intensity histograms. Combining HDIprep $\alpha$-MI extends image registration across technologies, without the need to match specific markers or analytes, and to embeddings that can differ in dimensionality, in an index-invariant manner. As part of the pipeline workflow, registration accuracy can and should be checked for suitability for the dataset by, e.g., manual inspection or by inspection of the Jacobian matrix of the spatial transformation. Since MIAAIM implements the Elastix toolbox, users can choose a variety of warping and interpolations methods that can be adapted according to the nature of the tissues requiring alignment (S2 Note). A detailed comparison of MIAAIM to alternative image registration methods is given in S2 Table. We used an original dataset of 30 prostate cancer (PC) biopsy images to directly compare MIAAIM registration to conventional manual landmarking registration with Elastix by quantifying the relative $\alpha$-MI gain between the IMC and MSI modalities after alignment with either method (S9 Fig). MIAAIM alignment performed similarly to conventional methods in 16/30 of samples (within $\pm 6\%$ gain or reduction in $\alpha$-MI). Of the remaining 14/30 samples, MIAAIM provided a 70–95% gain in 10 samples, with ~16–70% gain for 4 samples. These results demonstrate that MIAAIM registration offers overall performance improvement over conventional registration methods in approximately 50% of images investigated without incurring a clear reduction in performance across all images within the dataset. Further results from analysis of MIAAIM registered multimodal imaging of PC biopsies are presented in section "**MIAAIM validation on a multimodal prostate cancer TMA**" below.

## MIAAIM generates information on cellular phenotype, molecular ion distribution, and tissue state across imaging modalities on a Diabetic Food Ulcer (DFU)

To demonstrate the capacity of MIAAIM to generate linked data between multimodal imaging with various scales and data densities, it was applied to Matrix-assisted laser desorption/ionization time-of-flight (MALDI-TOF) MSI, H&E and IMC data from a diabetic foot ulcer (DFU) tissue biopsy containing a spectrum of tissue states, from the necrotic center of the DFU to the healthy margin. Image acquisition covered 1.2 cm$^2$ for H&E and MSI data. Molecular imaging with MSI enabled untargeted mapping of lipids and small metabolites in the 400–1000 mass-to-charge (m/z) range across the specimen at a resolution of 50 $\mu$m/pixel. Tissue morphology was captured with H&E at 0.2 $\mu$m/pixel on the same section, while 27-plex IMC data was acquired at 1 $\mu$m/pixel resolution from 7 ROIs on an adjacent section.

Cross-modality alignment was performed in a global-to-local fashion (**Fig 2c**). We aligned full-tissue data from MSI, H&E (down-sampled to approximately 3.5 $\mu$m$^2$/pixel), and a pre-IMC reference image (using a histological counterstain) first. Deformations not captured at the full-tissue scale within each IMC ROI were corrected using manual landmark guidance. Tissue deformations resulting from serial sectioning were accounted for with nonlinear transformations. Registrations were initialized by affine transformations for coarse alignment before nonlinear correction. Resolution differences were accounted for with a multiresolution smoothing scheme [28]. Final alignment proceeded by composing modality and ROI-specific transformations.

After locating individual cells using image segmentation, single-cell parameter quantification, and antibody staining quality control, registered images yielded the following information for 7,114 cells: (i) average expression of 14 proteins (**Methods, Table 1**), including markers for lymphocytes, macrophages, fibroblasts, keratinocytes, and endothelial cells, as well as extracellular matrix proteins, such as collagen and smooth muscle actin; (ii) morphological features, such as cell eccentricity, solidity, extent, and area, spatial positioning of each cell centroid; and (iii) the distribution of 9,753 m/z MSI peaks collected at a resolution of 50 µm$^2$/pixel across the full tissue. Distances from each MSI pixel and IMC ROI to the center of the ulcer, identified by manual inspection of H&E, were also quantified. Through the integration of these modalities, MIAAIM provided cross-modal information that could not be gathered with a single imaging system, such as the profiling of single-cell protein expression and microenvironmental molecular abundance.

## Identification of molecular microenvironmental niches correlates with cell and disease states

There is still little information available on immune niches and metabolic states within diabetic ulcers, although recent work has used high-dimensional methods such as spatial transcriptomics to identify cell types critically involved in tissue repair [44,45]. We verified the presence of cross-modal associations from the integrated multimodal imaging data set using *microenvironmental correlation network analysis* (MCNA) on registered IMC and MSI data (**Fig 3**). By performing community detection (i.e., clustering) on MSI analytes (m/z peaks) based on their correlations to single-cell protein measures, we defined *microenvironmental correlation network modules* (MCNMs; different colors in **Fig 3a**). Inspection of MCNMs with top correlations to protein levels identified with IMC revealed that sets of molecules, rather than individual peaks, were associated with cellular protein expression (**Fig 3b**). MCNMs organized on an axis separating those with moderate positive correlations to cell markers indicative of inflammation and cell death (CD68, cleaved caspase-3) from those with moderate positive correlations to markers of immune regulation (CD163, CD4, FOXP3) and vasculature (CD31). This strategy may be useful to detect molecular or metabolic states of migratory cells localized in areas of tissue with distinct microenvironments. Some proteins, such as the myeloid marker CD14 and the cell proliferation marker Ki-67, were not strongly correlated with any of the detected m/z peaks in the small molecule and lipid analysis across all cells.

To gain insights into the association of molecular distributions with tissue health, we plotted ion intensity distributions of MCNMs as a function of their proximity to the center of the ulcer (**Fig 3c**). This analysis revealed a change in molecular profiles about 6 mm from the ulcer center point, as the tissue state progressed from healthy to injured. We validated our

**Table 1. Key resource table.**

| Reagent or Resource | Source | Identifier |
|---|---|---|
| *Biological Samples* | | |
| **Diabetic Foot Ulcer IMC/MSI** | Fresh operative tissue sample at BIDMC | N/A |
| **Prostate Cancer MSI** | Fresh operative tissue sample at MGH | N/A |
| **Tonsil MSI** | Fresh operative tissue sample at MGH | N/A |
| **Prostate Cancer Tissue Microarray** | Tissue Biopsy Samples at MGH | N/A |
| *Critical Software/Algorithms* | | |
| **MIAAIM** | This study | https://github.com/MGH-VIC/miaaim-python-docker and https://mgh-vic.github.io/ |
| **Elastix** | Klein and Staring, et al.[28] | https://github.com/SuperElastix/elastix |
| **Docker** | Merkel [32] | https://www.docker.com |
| **MCMICRO** | Schapiro et al.[22] | https://mcmicro.org |
| **SCiLS Lab (2018)** | Bruker | https://scils.de |
| **Ilastik** | Berg et al.[30] | https://www.ilastik.org |
| **CellProfiler** | McQuin et al.[31] | https://cellprofiler.org |
| **Leiden Algorithm** | Traag et al.[75] | https://leidenalg.readthedocs.io/en/stable/intro.html |
| **Scipy** | Virtanen et al.[76] | https://www.scipy.org/index.html |
| **UMAP** | McInnes et al.[34] | https://umap-learn.readthedocs.io/en/latest/ |
| **Scikit-learn** | Pedregosa et al.[77] | https://scikit-learn.org/stable/ |
| **Scikit-image** | van der Walt et al.[78] | https://scikit-image.org |
| **Scikit-network** | Bonald et al.[79] | https://scikit-network.readthedocs.io/en/latest/ |
| **DFU IMC** | | |

| Target | Channel | Clone | Vendor | Catalog # |
|---|---|---|---|---|
| SMA | 141PR | 1A4 | Standard BioTools | 3141017D |
| CD19 | 142Nd | 6OMP31 | Standard BioTools | 3142014D |
| CD117 | 143Nd | 104D2 | Standard BioTools | 3143001B |
| CD14 | 144Nd | EPR3653 | Standard BioTools | 3144025D |
| CD16 | 146Nd | EPR16784 | Standard BioTools | 3146020D |
| CD163 | 147Sm | EDHu-1 | Standard BioTools | 3147021D |
| CD11b | 149Sm | EPR1344 | Standard BioTools | 3149028D |
| CD86 | 150Nd | IT2.2 | Standard BioTools | 3150020B |
| CD31/PECAM-1 | 151 Eu | EPR3094 | Standard BioTools | 3151025D |
| CD45 | 152Sm | CD45-2B11 | Standard BioTools | 3152016D |
| CD44 | 153Eu | IM7 | Standard BioTools | 3153029D |
| CD11c | 154Sm | Bu15 | Standard BioTools | 3154025D |
| FOXP3 | 155Gd | 236A/E7 | Standard BioTools | 3155016D |
| CD4 | 156Gd | EPR6855 | Standard BioTools | 3156033D |
| CD73 | 158Gd | EPR6115 | Standard BioTools | 3158031D |
| CD68 | 159Tb | KP1 | Standard BioTools | 3159035D |
| CD20 | 161Dy | H1 | Standard BioTools | 3161029D |
| Pan-Keratin | 162Dy | C11 | Standard BioTools | 3162033D |
| TGF-β | 163Dy | TW4-6H10 | Standard BioTools | 3163010B |
| Arginase-1 | 164Dy | D4E3M | Standard BioTools | 3164027D |
| β-Catenin | 165Ho | D13A1 | Standard BioTools | 3165032D |
| IL-10 | 166Er | JES3-9D7 | Standard BioTools | 3166008B |
| Ki-67 | 168Er | B56 | Standard BioTools | 3168022D |
| Collagen type 1 | 169Tm | Polyclonal | Standard BioTools | 3169023D |

*(Continued)*

**Table 1.** (Continued)

| Reagent or Resource | Source | Identifier | | |
|---|---|---|---|---|
| CD3 | 170Er | Polyclonal, C-Term | Standard BioTools | 3170019D |
| Caspase-3 | 172Yb | 5A1E | Standard BioTools | 3172027D |
| Pan-Actin | 175Lu | D18C11 | Standard BioTools | 3175032D |
| Histone-3 | 176Yb | D1H2 | Standard BioTools | 3176023D |
| **PROSTATE TMA IMC** | | | | |
| **Target** | **Channel** | **Clone** | **Vendor** | **Catalog #** |
| CXCR3 | 89Y | G025H7 | Biolegend | 353702 |
| Collagen type 1 | 93Nb | 3G3 | Santa Cruz Biotechnology | SC-293182 |
| p504s (AMACR) | 113In | 13H4 | Abeomics | 36-1144 |
| CD74 | 115In | LN2 | Biolegend | 326802 |
| CD38 | 141PR | EPR4106 | Standard BioTools | 3141018D |
| ERG | 142Nd | EPR3864 [2] | Abcam | AB133264 |
| CD39 | 143Nd | A1 | Biolegend | 328221 |
| CD11c | 144Nd | EP1347Y | Ionpath | 22084-06 |
| P63 | 145Nd | D-9 | Santa Cruz Biotechnology | SC-25268 |
| CD16 | 146Nd | EPR16784 | Standard BioTools | 3146020D |
| CD163 | 147Sm | EDHu-1 | Standard BioTools | 3147021D |
| CD45RA | 148Nd | HI100 | Biolegend | 304102 |
| CD15 | 149Sm | W6D3 | Biolegend | 2111918-17 |
| PTEN | 150Nd | 4C11A11 | Biolegend | 655002 |
| CD56 | 151Eu | MRQ-42 | Ionpath | 22062-01 |
| CD45 | 152Sm | D9M8I | Standard BioTools | 3152018D |
| CD44 | 153Eu | IM7 | Standard BioTools | 3153029D |
| CD14 | 154Sm | D7A2T | Ionpath | 19235-12 |
| IDO | 155Gd | D5J4E | Standard BioTools | 3155017D |
| CD4 | 156Gd | EPR6855 | Standard BioTools | 3156033D |
| CD11b | 157Gd | EPR1344 | Abcam | AB209970 |
| Keratin 34βE12 | 158Gd | 34βE12 | Biolegend | 916504 |
| CD68 | 159Tb | KP1 | Standard BioTools | 3159035D |
| CCR6 | 160Gd | G034E3 | Biolegend | 353402 |
| CD20 | 161Dy | H1 | Standard BioTools | 3161029D |
| CD8 | 162Dy | D8A8Y | Standard BioTools | 3162035D |
| CCR7 | 163Dy | G043H7 | Biolegend | 353202 |
| c-Myc | 164Dy | 9E10 | Standard BioTools | 3164025D |
| NKX3.1 | 165Ho | EPR14970 | Abcam | AB186413 |
| MBD1 | 166Er | 100B272.1 | Novus | NB100–56537 |
| EFCAB4B | 167Er | Rab Poly-IgG | Novus | NBP2–92725 |
| Ki-67 | 168Er | B56 | Standard BioTools | 3168022D |
| CD206 | 169Tm | E2L9N | Cell Signaling Technology | 49243SF |
| CD3 | 170Er | Polyclonal | Standard BioTools | 3170019D |
| HLA-DR | 171Yb | LN3 | Biolegend | 327002 |
| DCR3 | 172Yb | F-4 | Santa Cruz Biotechnology | SC-365755 |
| CD45RO | 173Yb | UCHL1 | Standard BioTools | 3173016D |
| CD7 | 174Yb | E4G1Q | Cell Signaling Technology | 32814BF |
| CD57 | 175Lu | HNK-1 | Biolegend | 359602 |

*(Continued)*

**Table 1.** (Continued)

| Reagent or Resource | Source | Identifier | | |
|---|---|---|---|---|
| ArrB2 | 176Yb | H-9 | Santa Cruz Biotechnology | SC-13140 |
| SMA | 181Ta | 1A4 | Santa Cruz Biotechnology | SC-32251 |
| E-Cadherin | 194Pt | 24 E10 | Cell Signaling Technology | 3195BF |
| CD10 | 195Pt | F-4 | Santa Cruz Biotechnology | SC-46656 |
| VCL | 196Pt | H-10 | Santa Cruz Biotechnology | SC-25336 |
| Keratin 8/18 | 198Pt | C51 | Santa Cruz Biotechnology | SC-8020 |
| NaK ATPase | 209Bi | F-2 | Santa Cruz Biotechnology | SC-514614 |

observations and the performance of HDIreg to align micron-scale structures by visualizing the distribution of top correlated ions within cellular microenvironments (**Fig 3d** and **3e**).

An advantage of the MIAAIM approach is the capacity to identify molecular variations in the MSI modality that correlate with cell states determined using the higher resolution and species-specific IMC modality. To investigate whether $m/z$ peaks differentially associate with cell proliferation (Ki-67 marker in IMC), we first identified cell phenotypes through unsupervised clustering on IMC segmented cell-level expression patterns (**Fig 3f**). This led to a differential correlation analysis between phenotypes within the well-separated CD3 + cell cluster, which identified infiltrating T cells at the wound site, as well as CD3- cell populations (**Fig 3g**). Interestingly, we found that correlations to Ki-67 expression shifted with near significance ($2\sigma$) for multiple $m/z$ peaks (Fisher transformed, one-sided z-statistics; Bonferroni corrected P-values) between CD3 − populations and the CD3 + population (**Fig 3h**), suggesting molecular candidates associated with proliferation in these cells. While the MALDI-TOF acquisition method used here does allow for putative assignment of analytes to a class, the instrument is not sufficiently accurate to allow precise identification of detected analytes. Further in-depth analysis using Fourier-transform ion cyclotron resonance (FT-ICR) mass spectrometry or Liquid Chromatography-Tandem Mass Spectrometry (LC-MS/MS) would enable analyte identification [46]. Interestingly, $m/z$ 820.35, likely belonging to the glycero-phosphoserine class [47] was closely associated with infiltrating CD3 + T cells at the injury site (**Fig 3d**). Functional associations are also enabled by the multimodal assessment of IMC and MSI analytes. For example, $m/z$ 550.82, likely a member of the ceramide class [47] was found to be negatively associated with T cell proliferation (**Fig 3h**). This is in line with a known role of ceramide in inducing apoptosis and reducing cell proliferation [48].

Utilizing the spatial context preserved by the MIAAIM workflow, we observed that ion intensities of $m/z$ peaks positively correlated to Ki-67 in CD3 + cells increased with distance from the wound, while molecules with Ki-67 negative correlations specific to CD3 + cells showed the opposite trend (**Fig 3i**). This distribution of Ki-67 correlates suggests that proliferation of CD3 + T cells occurs predominantly near the healthy margin of the DFU and confirms that molecular correlates of T cell proliferation can be identified through this unbiased analysis. Collectively, these results begin to provide insights into molecular microenvironments associated with different functional and metabolic states of cell subtypes, and how these microenvironments are distributed in the spatial context in a gradient from injured to healthy tissue in a diabetic foot ulcer.

### MIAAIM validation on a multimodal prostate cancer TMA

The robustness of MIAAIM was evaluated by applying it to IMC, MSI, and H&E imaging data from a prostate cancer tissue microarray (n = 30 cores, 90 total images). The TMA included samples from a range of histopathological grades, ranging from benign to Gleason scores of 4 + 5, as identified by a clinical pathologist's evaluation of H&E-stained sections (**Fig 4a**). Gleason scoring of histopathological prostate biopsy samples is based on tissue architecture and cellular organization, to describe normal (grade 1) to highly dysmorphic (grade 5), with the score determined by the two most predominant grades

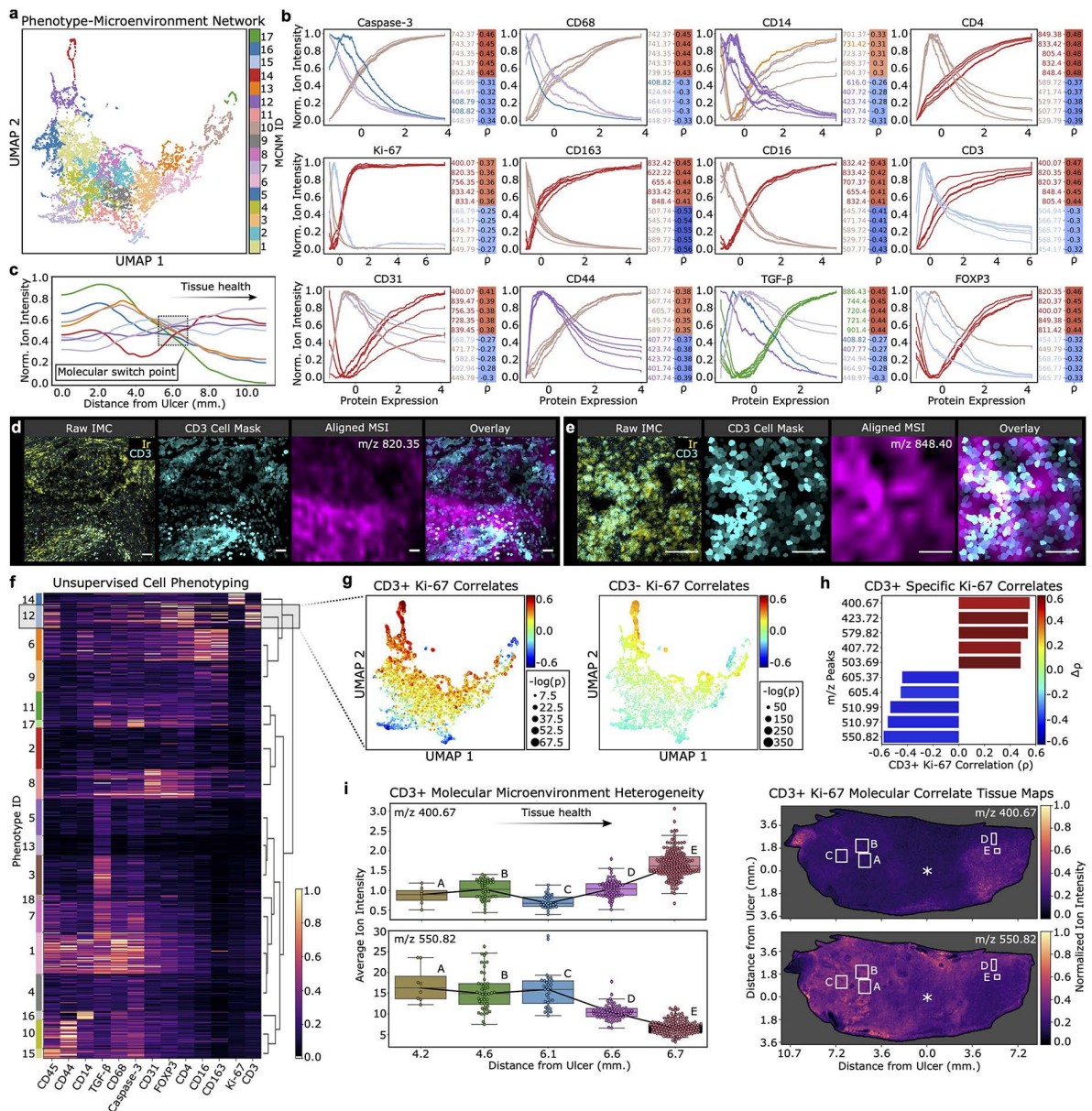

**Fig 3. Microenvironmental correlation network analysis (MCNA) links protein expression with molecular distributions in the DFU niche. a.** MCNA UMAP of *m/z* peaks grouped into modules. **b.** Exponential-weighted moving averages of normalized ion intensities (interpolated from alignment) for top five positive and negative correlates to proteins. Colors indicate module assignment. Heatmaps (right) indicate Spearman's rho. **c.** Exponential-weighted moving averages of normalized average ion intensity per modules ordered as distance from center of wound in DFU increases. **d.** Raw IMC nuclear (Ir) and CD3 staining in ROI (left) (scale bars = 80 μm). Masks showing CD3 expression (middle-left). Aligned MSI showing one of top CD3 cor-relates (middle-right). Overlay of CD3 expression and a top molecular correlate (right, interpolated MSI data). **e.** Same as **d** at different ROI. **f.** Unsuper-vised phenotyping. Shaded box indicates CD3 + population. Heatmap indicates normalized protein expression. **g.** MCNA UMAP colored to reflect ions' correlations to Ki-67 within CD3+ and CD3- populations. Colors indicate Spearman's rho and size of points indicates negative log transformed, Benjamini-Hochberg corrected P-values for correlations. **h.** Tornado plot showing top five CD3 + differential negative and positive correlates to Ki-67 compared to the CD3- cell populations. X-axis indicates CD3 + specific Ki-67 values. Color of each bar indicates change in correlation from CD3- to CD3 + populations. **i.** Boxplots showing ion intensity and of top differentially correlated ions (top, positive; bottom; negative) to CD3 + specific Ki-67 expression across ROIs on the DFU. Tissue maps of top differentially associated CD3 + Ki-67 correlates (top, positive; bottom; negative) with boxes (white) indicating ROIs on the tissue that contain CD3 + cells.

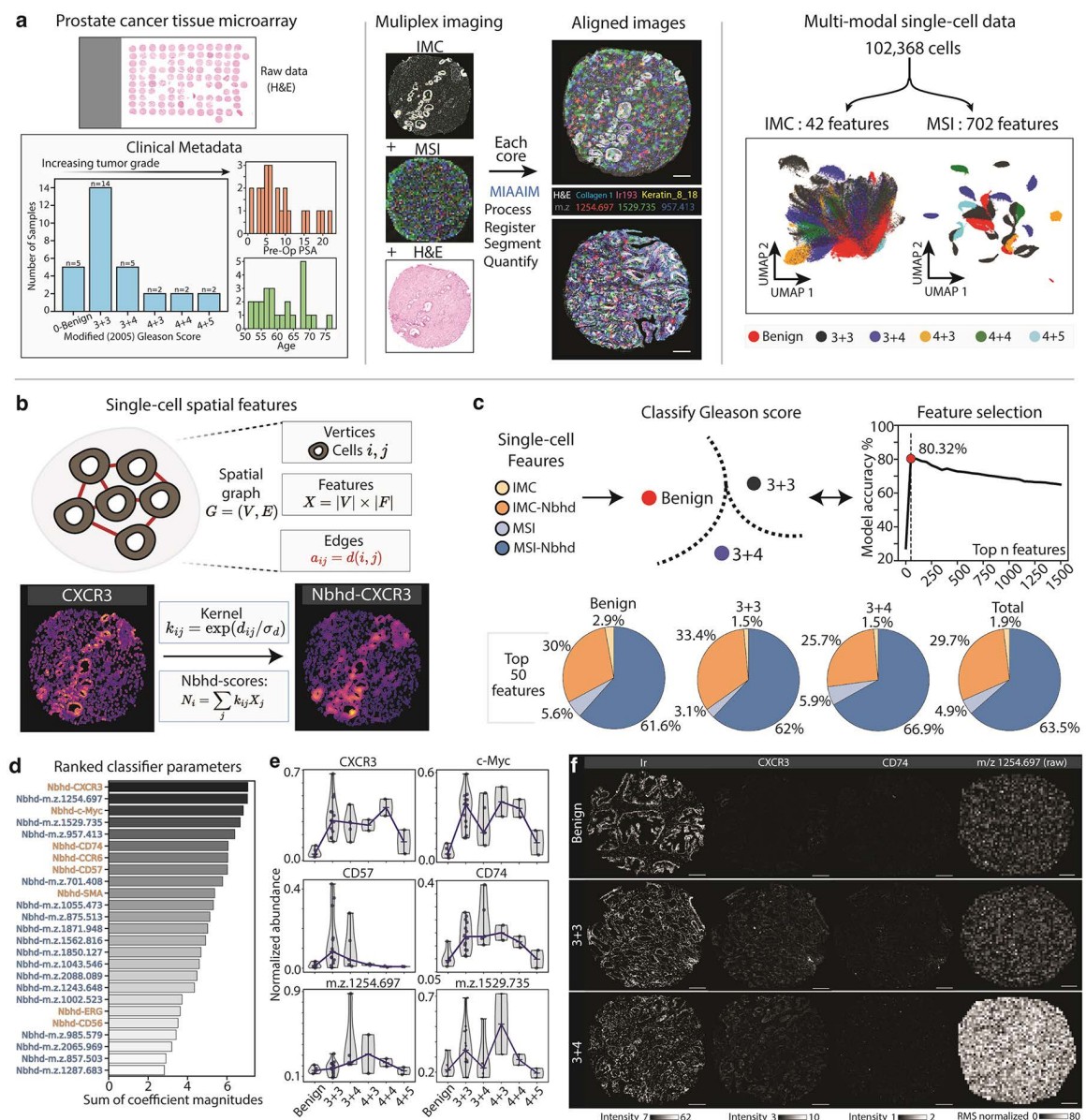

**Fig 4. Application of MIAAIM to prostate cancer TMA generates signatures of Gleason score. a**. MIAAIM applied to MSI, H&E, and IMC data from prostate cancer TMA composed of cores ranging from Gleason scores of 0-Benign to 4+5, with majority of samples having scores of 3+3 (scale bars = 150 μm). Compositional alignment applied to each biopsy produced overlayed modalities for 30 samples (interpolated MSI and H&E shown). Cell segmentation and quantification of multimodal single-cell data shows TMA-specific signatures (UMAP embeddings) **b**. Neighborhood interaction scores are calculated for each cell to obtain continuous spatial features for each modality. **c.** Model to classify between benign, 3+3 and 3+4 Gleason scores. After establishing an initial feature importance ranking, we ran models with the top n models to determine that 50 features are sufficient to generate a high accuracy model. Examining these top 50 features reveals that most of the strong classification features are spatial neighborhood signals across each classification model and overall. **d.** While MSI features make up the majority of the top 50 features, 6 of the top 10 features are IMC features, specifically CXCR3 and c-Myc. **e.** Examining single cell signals of TMAs of each Gleason score class under each modality shows discernible intensity variation between the Gleason score classes in the CXCR3 and *m/z* 1529.735 parameters.

[49]. The varying Gleason scores and the high epithelial surface area of prostate tissue contained in the TMA presented challenges for image alignment, beyond high dimensionality, relative to the DFU dataset.

Untargeted analysis of the tumor microenvironment was performed using MSI after tryptic digestion, to map protein peptides ranging between 500–2500 $m/z$ at a resolution of 20 $\mu m^2$/pixel. Subsequently, H&E staining and imaging at 0.5 $\mu$m/pixel was performed on the same section as used for MSI. An adjacent section was labeled with a 46-plex antibody panel to generate IMC data (1 $\mu m^2$/pixel resolution) (**Methods, Table 1**). An adjacent section was labeled with a 46-plex antibody panel to generate IMC data (1 $\mu m^2$/pixel resolution) (**Table 1**). The compositional alignment was applied for each core using the approach from the DFU analysis, resulting in 60 separate image registration tasks (2 per core). MIAAIM produced both visually accurate alignments with quantifiable improvements over manual registration, both globally and within smaller-scale cellular structures, for each histopathological grade and modality (**Figs 4a** and S9). Importantly, registration performance gains using MIAAIM relative to conventional manual landmarking approaches were observed across tissue states (i.e., Gleason grades). This indicates that MIAAIM performance gains were not tied to gross changes in tissue architecture associated with disease progression.

## MIAAIM enables multimodal classification of prostate cancer histopathological grade

After assessing the robustness of MIAAIM registration, we evaluated the utility of combined multimodal IMC and MSI data generated by MIAAIM by quantifying the relative importance of each modality and corresponding spatial features to predict Gleason score utilizing single-cell information. Paired IMC/MSI profiles for approximately 100,000 cells were generated using single-cell segmentation masks from the IMC modality to enable quantification of average multimodal features per cell. Unlike single-cell features, such as average expression, marker-wise spatial features are not directly computable after cell segmentation, and often spatial analyses require partitioning cells into discrete types either by manual annotation or cluster analysis [6,29,50]. To obtain continuous spatial features for each cell, we calculated neighborhood "interaction" scores, akin to Gaussian filters in image processing (**Fig 4b**). These scores derive from spatial neighborhood graphs built from segmented cells in a sample, where graph nodes are cells, and edges connect cells within a specified radius of each other. The interaction score of a cell captures the aggregate signal within its neighborhood and is computed as the sum of the cellular signal of each of its neighbors, weighted by a distance-scaled Gaussian kernel built from graph edges.

Interaction scores for each cell and biopsy were computed using IMC and MSI single-cell data and a radius of 75 $\mu$m. Logistic regression was then performed on (i) individual modality datasets, (ii) augmented single modalities (e.g., MSI data with single cell segmentation information), and (iii) all features (paired IMC + IMC interaction, and MSI + MSI interaction), to assign a Gleason score to each cell (**Fig 4c**). Model cross-validation was performed per biopsy, rather than per cell. Due to low sample sizes for high-grade biopsies, only 3 classes were used (Benign, 3 + 3 and 3 + 4). Models incorporating spatial interaction scores reached class-balanced accuracies 1–3% higher than models using only single-cell features on test sets. Additionally, IMC data outperformed MSI data in classification by 4–6%. Incorporating all four sets of features – IMC, IMC interaction scores, MSI, and MSI interaction scores resulted in the best accuracy, which highlights an advantage of integrated multimodal imaging data for more complete tissue representations compared to individual modalities alone and over multimodal modalities without spatial context (S3 Table).

Classification accuracy was refined by iteratively training models using the top $n$ features from the model incorporating all features (sorted by decreasing absolute value of model coefficients) (**Fig 4c**). The top 50 predictors produced a model with a 3-fold cross validation balanced accuracy of ~80%, with over 90% of classification signal sourced from spatial features. These patterns agree with the visual evaluation of tissue pathology in practice. Interaction scores represent signals in cellular neighborhoods that more closely mimic the physical scale that Gleason scores are assigned at when combined as a representation of tissue state, while single-cell measures represent localized phenotypic state.

While the majority (>60%) of predictors were MSI interaction features (**Fig 4d**), IMC interaction scores were the most highly represented among the top 50 and top 25 predictors (~10 times more likely than chance) (**Fig 4d**). The top IMC and

MSI spatial features were the chemokine receptor CXCR3 and *m/z* 1254.697, respectively. Altered expression CXCR3 is linked to PC progression and metastasis [51]. While the MALDI TIMS-TOF instrument utilized here lacks the ability to provide definitive analyte identification, a previous study showed that *m/z* 1562.795 (analogous to *m/z* 1562.816 in our study) is among the most prominent peaks in trypsin-digested prostate cancer tissue imaged by MALDI-MSI and subsequently determined by LC-MS/MS that it is derived from collagen α-2 (I) (Uniprot ID: P08123) [11]. Thus, these results demonstrate both the importance of integrated spatial analyses and a first approximation of estimating feature significance to inform data interpretation and study design.

Various pre-existing models use deep learning to assign Gleason scores to H&E-stained tissues [52,53]. In contrast, multimodal features generated by MIAAIM enable a simple model at single-cell resolution, achieving comparable accuracy on this limited dataset, along with the ability to identify explanatory molecular features. The proto-oncogene c-Myc and CXCR3 are known biomarkers of prostate cancer tumorigenesis and progression [54–57], and were top IMC predictors, showing an increasing trend in expression as a function of Gleason score up to the 4 + 5 class (**Fig 4e**). A similar trend was observed to the 4 + 4 class for the membrane receptor CD74, whose ligand, macrophage migration inhibitory factor (MIF), is expressed at elevated levels in prostate cancer cells [58,59]. However, the mechanism by which CD74 regulates prostate cancer cell viability is less clear [60]. Previous studies also show that MSI detection of glycans, lipids and metabolites can distinguish between benign and cancerous prostate tissue [61–64]. Similarly, our results indicate that MSI analysis of protein-derived peptides can identify discriminatory signatures of prostate carcinoma.

In line with previous models [52,53], classification of benign versus cancerous samples achieved a higher accuracy task than distinguishing between Gleason classes (S10 Fig). Unregistered (i.e., raw) IMC and MSI data agreed with our model's output, indicating consistency between original measurements and augmented data from registration and cell segmentation with MIAAIM (**Fig 4f**). Collectively, these results demonstrate a clear advantage of integrated multimodal data in a spatial context. Additionally, we show that MIAAIM is robust to complex tissue types, and that it may be a resource to generate multimodal, single-cell and pixel-level data for the identification of biomarkers related to diseased tissue state in practical application, given larger sample sizes.

In summary, the analysis of prostate cancer TMA images demonstrates that MIAAIM can generate multimodal integrated single-cell and pixel-level spatial data which can be used to interrogate the cellular and biomolecular architecture and help identify signatures that confidently discriminate between disease-related tissue states.

## Discussion

We present MIAAIM, a reproducible tissue and disease agnostic framework to align data from separate multiplexed imaging modalities and convert these data into multimodal single-cell measurements to discover and provide insights into novel pathologic mechanisms and therapeutic targets along with diagnostic and prognostic indicators to support clinical decision making. We applied MIAAIM to combine IMC, MSI, and H&E imaging datasets, enabling the analysis of molecular microenvironmental niches of individual cells correlated with their states and localization within tissue sections in the diverse contexts of diabetic foot ulcer and prostate cancer biopsies. The robustness of MIAAIM was evaluated across healthy and pathological human tissue types and imaging technologies, along with the potential practical utility of using its output for identifying biomarkers of tissue state.

In contrast to multimodal registration methods that require images with low numbers of channels or datasets with prior knowledge of correspondences between imaging channels, MIAAIM is technology-agnostic and can align datasets without channel-to-channel matchings across modalities. This generalization is achieved by first estimating pixel embedding dimensionalities that maximize retained information for each modality. It then optimizes a spatial transformation to maximize the similarity of nearest-neighbor graph edge lengths across embeddings, rather than comparing pixel intensities (**Fig 2**). Previously, 3-D pixel embeddings have been used for multimodal registration but apply RGB-to-grayscale conversions, which constrains the information content retained in embeddings and imposes additional bias through grayscale

conversion [42,43]. MIAAIM also implements subsampling and spectral landmarking to optimize computational efficiency of pixel embeddings, and it uses Elastix [28] to find image-to-image transformations, which allows the use of many existing approaches to align low-channel images, such as immunohistochemical staining, or other modalities with corresponding reference stains.

We evaluated MIAAIM on a total of 111 images across three different imaging technologies (MSI, IMC, and H&E) and two human tissue types, including whole-tissue MSI and H&E data from a DFU biopsy with corresponding IMC data of selected ROIs on an adjacent section as well as a prostate cancer TMA (**Figs 3** and **4**, respectively). We identified molecular correlates of T-cell proliferation in the DFU through cross-modal IMC and MSI analyses and tracked their abundance as tissue transitioned from wound bed to healthy. The analysis of the DFU biopsy tissue demonstrated that molecular microenvironments of cell subtypes associate with their functional and metabolic state, and that these microenvironments distribute in a continuous gradient from injured to healthy tissue. These results demonstrate MIAAIM's potential to provide data which connects biological scales – from cell state to local molecular niches, to tissue state – across biomolecular classes.

The outputs of MIAAIM can be used to build interpretable models from a variety of imaging datasets. Harnessing the co-registered IMC and MSI data from 30 prostate biopsies, we built a classification model that identified signatures of prostate disease according to their histopathological score (Gleason Score - benign vs. 3 + 3 vs. 3 + 4). Importantly, the classification signals driving the best performing model for PC biopsy scoring are dominated by spatial information forms of IMC and MSI features. To determine core drivers of classification model performance, a feature selection approach identified both MSI and IMC analytes as key classifiers (e.g., $m/z$ 1254.697, CXCR3, and cMYC). In agreement with previous reports, $m/z$ 1562.816 is associated with collagen within a prostate tissue signature [11]. Similarly, cMYC, CD74, and CXCR3 were important to model performance and are linked to prostate cancer aggressiveness.

Identification of known prostate cancer biomarkers together with spatial components highlights a key advantage of *in situ* alignment of these paired modalities and multimodal image integration to build more complete tissue state representations, as compared to individual imaging modalities or multimodal datasets without spatial context. Precisely characterizing protein fragments without direct chemical identification by methods such as LC-MS/MS is difficult due to the lack of a reference library based on proteomic MALDI-MSI [11], however, identification is not required for classification, and once molecular candidates with high predictive power are found, they can be further identified using tandem MS (MS/MS) fragmentation. For MSI analytes identified either from proteomic reference libraries or by MS/MS analysis, cognate antibodies suitable for immunolabeling can be utilized to provide orthogonal validation. Similarly, IMC analytes with high-predictive utility or biological interest can be assessed through assessment by immunofluorescence or immunohistochemistry. These validation approaches, together with support from published reports, can confirm biological associations initially identified through analysis of multi-modal high-parameter images.

Here, we have demonstrated the utility of MIAAIM as a tool for detecting novel associations and predictors of cell and tissue state in an unbiased manner. Interrogation of the complex and high dimensional data produced by MIAAIM image integration requires strategies to limit spurious associations and reduce model overfitting. For example, in the analysis of the DFU data, false discovery rate control (Benjamini-Hochberg correction) was used to reduce the number of MSI peaks that significantly correlate to cellular substates within a CD3 + cell population. In the context of the prostate cancer TMA dataset, regularization and iterative refinement of parameter sets were used to shrink and remove coefficients of uninformative parameters to find the 50 most performant features for classification. Incorporation of these statistical procedures can increase the confidence of the identified associations. However, definitive studies on larger and separate sample sets are necessary to evaluate candidate biomarkers of disease or clinical state.

As with other nonparametric image alignment methods, MIAAIM benefits from parameter-tuning for optimal performance, and often requires manual landmark guidance in complex tissue types. Although deep learning may have the potential to alleviate these challenges, currently there are no learning models for multiplexed image registration without

reference stains to our knowledge. Similarly, while there are several benchmarks for multimodal co-registration of low-plex modalities, such as H&E stains [65], there is a need for applications targeted to multiplexed modalities, since most current benchmarking metrics cannot be applied. MIAAIM provides parameter map logging for iterative workflow reproducibility that may also aid in generating standardized training sets for potential AI-based methods. Further, the modular implementation of MIAAIM enables the incorporation of new developments, such as advances from the deep learning community, which may facilitate future benchmarks.

In conclusion, MIAAIM provides a conceptual and practical method for novel applications in high-dimensional, multimodal bioimage integration. This framework is also technology-agnostic and accommodates any data in BioFormats/OME-compatible, imzML, NIfTI, and HDF5 formats. We envision MIAAIM to have applications ranging across imaging modalities including single-cell proteomic, transcriptomic, and molecular imaging. The harmonization of these data is well-positioned to provide more complete tissue portraits across health and disease states, and the transitions between them that will provide novel insight into disease mechanism, diagnosis, therapeutic targeting, prognosis and ultimately clinical decision making.

## Methods

### Ethics statement

All patient tissue samples were obtained with approval from the Institutional Review Boards (IRB) of Massachusetts General Hospital (protocol #2005P000774) and Beth Israel Deaconess Medical Center (protocol #2018P000581). Written informed consent was obtained from all study participants from both Beth Israel Deaconess Medical Center and Massachusetts General Hospital.

### Biological methods

**Imaging mass cytometry of full tissue samples.** Frozen tissues from a single DFU, tonsil, or prostate biopsy (n = 1 each) were sectioned serially at a thickness of 10 $\mu$m using a Microm HM550 cryostat (Thermo Scientific) and thaw-mounted onto SuperFrost™ Plus Gold charged microscopy slides (Fisher Scientific). After temperature equilibration to room temperature, tissue sections were fixed in 4% paraformaldehyde (Ted Pella) for 10 min, then rinsed 3 times with cytometry-grade phosphate-buffered saline (PBS) (Fluidigm). Unspecific binding sites were blocked using 5% bovine serum albumin (BSA) (Sigma Aldrich) in PBS including 0.3% Triton X-100 (Thermo Scientific) for 1 hour at room temperature. Metal conjugated primary antibodies (Fluidigm) at appropriately titrated concentrations were mixed in 0.5% BSA in DPBS and applied overnight at 4 °C in a humid chamber. Sections were then washed twice with PBS containing 0.1% Triton X-100 and counterstained with iridium (Ir) intercalator (Fluidigm) at 1:400 in PBS for 30 min at room temperature. Slides were rinsed in cytometry-grade water (Fluidigm) for 5 min and allowed to air dry. Data acquisition was performed using a Hyperion Imaging System (Fluidigm) and CyTOF Software (Fluidigm), in 33 channels, at a frequency of 200 pixels/second and with a spatial resolution of 1 $\mu$m. Images were visualized with MCD Viewer software (Fluidigm) before exporting the data as text files for further analysis. After imaging, slides were rapidly stained with 0.1% toluidine blue solution (Electron Microscopy Sciences) to reveal gross morphology. Slides were digitized at a resolution of approximately 2.75 $\mu$m/pixel using a digital camera.

**Generation of mass spectrometry imaging data from full tissue samples.** Paired 10 $\mu$m thick sections from the same tissue blocks from DFU, tonsil, or prostate biopsy (n = 1 each) used for imaging mass cytometry were thaw mounted onto Indium-Tin-Oxide (ITO) coated glass slides (Bruker Daltonics). Tissue sections were coated with 2.5-dihydroxybenzoic acid (40 mg/mL in 50:50 acetonitrile:water including 0.1% TFA) with an automated matrix applicator (TM-sprayer, HTX imaging). Mass spectrometry imaging of sections was performed using a rapifleX MALDI Tissuetyper (Bruker Daltonics, Billerica, MA). Data acquisition was performed using FlexControl software (Bruker Daltonics, Version 4.0) with the following parameters: positive ion polarity, mass scan range (*m/z*) of 300–1000, 1.25 GHz digitizer, 50 $\mu$m

spatial resolution, 100 shots per pixel, and 10 kHz laser frequency. Regions of interest for data acquisition were defined using FlexImaging software (Bruker Daltonics, version 5.0), and individual images were visualized using both FlexImaging and SciLS Lab (Bruker Daltonics). After data acquisition, sections were washed with PBS and subjected to standard hematoxylin and eosin histological staining followed by dehydration in graded alcohols and xylene. The stained tissue was digitized at a resolution of 0.5 $\mu$m/pixel using an Aperio ScanScope XT brightfield scanner (Leica Biosystems).

**DFU mass spectrometry imaging data preprocessing.** Data for the DFU were processed in SciLS LAB 2018 using total ion count normalization on the mean spectra and peak centroiding with an interval width of $\pm$25mDa. For all analyses, a peak range of *m/z* 400–1,000 was used after peak centroiding, which resulted in 9,753 *m/z* peaks. No peak-picking was performed for presented data unless stated otherwise. Data were exported from SciLS Lab as imzML files for further analysis and processing.

**Generation of prostate cancer TMA imaging data.** Serial 5-micron-thick FFPE sections were cut from a tissue microarray (TME) containing 1-millimeter cores of formalin-fixed- paraffin-embedded (FFPE) tissues of prostate cancer biopsy specimens. Sections were placed onto superfrost gold plus slides and stored under vacuum prior to processing.

**Generation of prostate TMA imaging mass cytometry data.** For imaging mass cytometry, slides were incubated for 18 hours at 60°C and cooled to room temperature. Next, wax was removed by two 3 min xylene incubations, followed by rehydration in ethanol baths (100% 1min, 100% 1min, 95% 1min, 70% 1min – diluted in ddH20). Antigen retrieval was performed using Ventana CC1 buffer in a Biocare DC2012 Decloaking chamber set to 95°C for 40 min. After cooling to 70°C, tissue was incubated in PBS for 5 min three times, followed by incubation in 3% BSA for 60 min at room temperature. Tissue sections were labeled with metal-conjugated antibodies overnight at 4°C. Following incubation, the sections were incubated at room temperature in PBS 0.2% Triton X-100 for 8 min twice, followed by wash in PBS for 8 min, and then incubated with Iridium Intercalator (1:3500) for 30 min. Following a wash for 10 sec in PBS, the slide was incubated in freshly prepared 0.00025% w/v solution of RuO4 for 3 min, and washed with ddH20 for 10 sec, prior to air drying. Individual ROIs were set for each of the 30 individual cores imaged from the TMA and acquired on a Standard Biotools Hyperion system at resolution of 1μm$^2$.

**Generation of prostate TMA mass spectrometry imaging data.** For MSI imaging samples, an adjacent FFPE tissue section was deparaffinized with xylene and subjected to antigen retrieval in 100 mM Tris at pH 9 for 20 min at 95 °C using a Biocare DC2012 Decloaking chamber (Biocare Medical, Pacheco, CA, USA). Using a HTX M5 Robotic sprayer, the slide was coated with trypsin and incubated for 4 h at 37 °C. Next, a matrix composed of 10 mg/mL α-cyano-4-hydroxycinnamic acid (CHCA) matrix in 70% ACN, 0.1% TFA, 10 mM ammonium phosphate was applied with the HTX Sprayer. ROIs corresponding to the same 30 biopsy cores imaged by IMC were selected and imaged using a Bruker timsTOF fleX MALDI QTOF Mass Spectrometer with a full mass spectrum collected at every position (pixel) over the surface of the tissue section at a resolution of spatial resolution 20μm$^2$. Centroid values were used to label m/z peaks.

**Prostate TMA imaging data preprocessing.** For imaging processing, multi-stack TIFF image files were exported using MCD Viewer software (Standard Biotools). To remove background and folded tissue, masks were generated in ImageJ. Extraneous channels were extracted from individual image stacks, reducing the number from 64 channels to 49 channels. To identify individual cells, images were segmented with random forest pixel classifier trained in Ilastik based on cytoplasmic channels (CD45, Calretinin) and a nuclear channel (Ir193) [66]. Probability maps from Ilastik output were saved as red, green and blue tiff files and fed into CellProfiler to generate whole-cell image masks.

No noise annotations or artifact removal steps were used. Two 250 $\mu$m by 250 $\mu$m cropped regions for each core were required to contain more than 5% of pixels over the previously defined threshold for training. For each core and modality, a mask outlining the contours of the image was created and transformed to the final aligned image stack. Segmentation masks from the MIAAIM CellProfiler module were obtained by identifying segmented cells contained in the intersection of these contour masks with the image masks and removing those cells touching the border.

**MIAAIM implementation.** MIAAIM workflows are implemented in a Docker-containerized Python package to enable reproducible workflows and eliminate any platform-specific dependencies. MIAAIM's output interfaces with existing image analysis software tools (see S1 Note, **Combining MIAAIM with existing bioimaging software**).

**High-dimensional image compression and pre-processing (HDIprep). The** HDIprep module is an image preprocessing pipeline to prepare multimodal images for registration. HDIprep includes dimensionality reduction for high-parameter data and filtering and morphological operations for single-channel images. Processed images were exported as 32-bit NIfTI-1 images using the NiBabel Python library. NIfTI-1 was chosen as the default file format for many of MIAAIM's operations due to its memory mapping feature in Python and compatibility with Elastix and ImageJ.

To reduce the dimensionality of high-parameter images, HDIprep estimates an optimal UMAP embedding dimensionality for pixel-level data. Dimension reduction is initialized with optional, spatially-guided subsampling to reduce data set size (see below). Uniform Manifold Approximation and Projection (UMAP) is then used to construct a graph representing the data manifold and its underlying topological structure [34]. UMAP aims to optimize a lower-dimensional image embedding of a high-dimensional image by generating a representative fuzzy simplicial set (i.e., a weighted, undirected graph), such that the fuzzy set cross-entropy between the embedded simplicial set and the high-dimensional counterpart is minimized.

Calculating cross-entropy scales quadratically with the number of data points, making its use for large data sets impractical. In contrast, UMAP does not compute the exact cross entropy when optimizing low-dimensional embeddings. Instead, it uses probabilistic edge sampling and negative sampling to reduce runtimes for large data sets [34]. To ensure accurate estimates of the embedding error, we compute landmarks on the data manifold that are representative of its global structure, and we use these in the calculation of the exact cross-entropy over a range of dimensionalities.

To identify representative landmarks on the data manifold, we use a variant of spectral clustering on the fuzzy simplicial set (detailed below). We iteratively project the spectral centroids into Euclidean spaces of increasing dimensionality using UMAP, and calculate the exact fuzzy set cross-entropy in each. To estimate the optimal embedding dimensionality, a least-squares exponential regression is fitted to the min-max normalized cross-entropy as a function of dimensionality; samples are then simulated along the regression line to find the first dimensionality falling within the 95% confidence interval of the exponential asymptote. The subsampled data is embedded into Euclidean space with this dimensionality, and out-of-sample pixels are projected into this embedding using the native nearest-neighbor based method in UMAP (*transform ()* function). Finally, all pixels are mapped back to their original spatial coordinates to construct a "*compressed*" image with the number of channels equal to the indicated embedding dimensionality. These steps are summarized in the following sections and in pseudo-code below:

**Algorithm 1: Image Compression.**

```
Input: Multichannel image (X), SVD dimensionality (b), k-means clusters (k), embedding dimensions
i = 1,..., n
Output: Compressed image (I)
function Compress
        {Subsample Pixels}
        p ⟵ ⊆{a_{x,y} | a_{x,y} ∈ X}
        {Compute Data Manifold}
        g ⟵ FuzzySimplicialSet (p) ▷ UMAP
        {Compute Spectral Centroids}
        s ⟵ RandomizedSVD (g, b)
        c ⟵ MiniBatchKmeans (s, k)
        {Compute Manifold Embedding Error}
        g_c ⟵ FuzzySimplicialSet (c)
        for i = 1...n do
                e_c^i ⟵ EmbedSimplicialSet (g_c, i)
                {Compute Fuzzy Set Cross Entropy}
                E_i ⟵ FuzzySetCrossEntropy (g_c, e_c^i)
```

```
E ⟵ (E−E_min)/(E_max−E_min)  ▷ Min-max Normalization
{Estimate Embedding Dimensionality}
ŷ ⟵ (y_0 − c) e^{−kE} + c  ▷ Exponential Regression
for i = 1...n do
        if ŷ_i in c ± 1.96σ do  ▷ Model Gaussian Residual Process
                j ⟵ i
                break
{Compute Pixel Embedding}
for k_{x,y} in { a_{x,y} | a_{x,y} ∈ X ∧ a_{x,y} ∉ p } do
        M ⟵ ProjectData (k_{x,y}, e_c^j) ▷ Project data (UMAP transform ())
{Reconstruct Image}
for k_{x,y} ∈ M do
I (x, y) ⟵ k_{x,y}
return I
```

Image data subsampling. To reduce computational burden during dimension reduction, subsampling can be performed at the pixel level; this is optional depending on image size and available computational resources. Subsampling options include uniformly spaced grids, random coordinate selection, and random selection initialized with uniformly spaced grids (pseudo-random). HDIprep also supports restricting these subsampling approaches to masked sampling regions, which may be useful for large data sets.

By default, images with fewer than 50,000 pixels are not subsampled, images with 50,000–100,000 pixels are subsampled using 55% pseudo-random sampling initialized with 2x2 pixel uniformly spaced grids, images with 100,000–150,000 pixels are subsampled using 15% pseudo-random sampling initialized with 3x3 pixel grids, and images with over 150,000 pixels are subsampled with 3x3 pixel grids. These default values are based on empirical studies (S4, S5, and S6).

No subsampling was used for the presented MSI data from the DFU. Subsampling rates for presented IMC data from the DFU were determined on a case-by-case basis from empirical studies, and match those used in the spectral landmark sampling experiments. Subsampling with 10x10 pixel uniformly spaced grids was used for CyCIF data. All IMC and MSI data from prostate cancer biopsies in the TMA dataset were processed using the default parameters.

Landmark selection with spectral clustering. Spectral landmarks are identified using a variant of spectral clustering. Specifically, randomized singular value decomposition (SVD) followed by mini-batch k-means is used to scale spectral clustering to large data sets, following the procedure introduced in the PHATE algorithm [67]. Given a symmetric adjacency matrix representing pairwise similarities between nodes (in this context, pixels), we compute the eigenvectors corresponding to the largest eigenvalues. Mini-batch k-means is then performed on the nodes using these eigenvectors as features. Spectral landmarks are defined as the centroids of the resulting clusters in the original space.

By default, input data is reduced to 100 components using randomized SVD, then split into 3,000 clusters using mini-batch k-means. These default values are based on empirical studies (S7 Fig). Due to the suggested dimensionalities of embeddings of MSI and IMC data only being available after experimental tests, no landmark selection was used for processing or determining the optimal embedding dimensionality of these data sets. Instead, full or subsampled data sets were used. All other embeddings employing the dimensionality estimate for image data used the above default parameters.

UMAP embedding dimensionalities. By default, HDIprep embeds spectral landmarks into Euclidean spaces with 1–10 dimensions to identify embedding dimensionalities suggested for UMAP. Exponential regressions on the spectral landmark fuzzy set cross entropy are performed using built-in functions from the SciPy Python library. These default parameters were used for all presented data.

Image compression with UMAP parametrized by a neuronal network. We implemented parametric UMAP[36] using the default parameters and neural architecture with a TensorFlow backend. The default architecture was comprised of a

3-layer 100-neuron fully connected neuronal network. Training was performed using gradient descent with a batch size of 1,000 edges and the Adam optimizer with a learning rate of 0.001 [68].

Histology image preprocessing. HDIprep processing options for low-channel histological stains include image filters (e.g., median), thresholding (e.g., manually set or automated), successive morphological operations (e.g., opening and closing), and masking. Presented H&E and toluidine-blue stained images were processed using median filters to remove salt-and-pepper noise and Otsu thresholding to create a binary mask for the foreground. Sequential morphological operations were then applied to this mask, including opening to remove small, connected foreground components, closing to fill small holes, and filling to close large holes.

**High-dimensional image registration (HDIreg).** HDIreg implements the open-source Elastix software [28] in conjunction with Python modules for the image resizing, padding, and trimming often applied before registration. Several different registration parameters, cost functions, and deformation models are included in Elastix; HDIreg additionally allows manual definition of point correspondences for problematic inputs, as well as composite transformations for fine-tuning (see S2 Note, **Notes on the HDIreg workflow's expected performance**).

High-parameter images are registered using a manifold alignment approach, which aims to maximize image similarity. Image registration is an optimization problem to determine the smooth transformation parameters which maximize the selected similarity measure (in this case, Rényi $\alpha$-mutual information) between the transformed target image and a fixed reference image.

Differential geometry and manifold learning. MIAAIM's default alignment approach uses the entropic graph-based Rényi $\alpha$-mutual information ($\alpha$-MI) [39] as similarity measure, which extends to manifold representations of images embedded in Euclidean space with potentially differing dimensionalities (e.g., UMAP pixel embeddings). This measure is justified through an intrinsic definition of the Rényi $\alpha$-entropy (S4 Note).

Briefly, given a Lebesgue density drawn from independent and identically distributed random vectors supported by a compact, n-dimensional Riemannian manifold, its extrinsic alpha-entropy can be approximated using continuous quasi-additive graphs, which includes k-nearest neighbor (KNN) Euclidean graphs [69], as their edge lengths asymptotically converge to the Rényi $\alpha$-entropy of feature distributions as the number of feature vectors increases [70].

This property leads to the convergence of KNN Euclidean edge lengths to the *extrinsic* Rényi $\alpha$-entropy of a set of random vectors with values in a compact subset of $\mathbb{R}^d$ (with $d \geq 2$) [37]. Costa and Hero [38] generalized these results to embedded manifolds by estimating an *intrinsic* Rényi $\alpha$-entropy and applied their framework along with Isomap and its variant C-Isomap for intrinsic dimensionality estimation.

In contrast to results that require all pairwise approximations for each point in the data set to estimate the $\alpha$-entropy, we aim to provide a similar formulation utilizing local information, following the results of our dimension reduction benchmark, which shows that local information preserving algorithms are well-suited for high-dimensional image data (see S1, S2, S3 Figs and S3 Note, **HDIprep dimension reduction validation**).

Entropic graph estimators on local information of embedded manifolds. Using the power-law relationship between volumes of open neighborhoods in UMAP and their embedded counterparts, we can use exponential regression to identify the dimensionality $m$ such that geodesics within open neighborhoods are preserved. KNN graph functionals calculated in this estimated embedding space provide the necessary machinery to calculate the intrinsic $\alpha$-entropy of embedded data manifolds in MIAAIM by applying the **Beardwood-Halton-Hammersley Theorem** using the induced volume measure across all coordinate patches (S4 Note, BHH).

Rényi $\alpha$-MI provides a quantitative measure of association between the intrinsic structure of multiple manifold embeddings constructed with the UMAP algorithm. The Rényi $\alpha$-MI measure extends to feature spaces of arbitrary dimensionality, which MIAAIM utilizes in combination with its image compression method to quantify similarity between optimized embeddings of image pixels in potentially differing dimensionalities.

We note that the *information density* of volumes of continuous regions of model families (i.e., collections of output embedding spaces or input points) have been recognized in defining an information geometry of statistical manifold learning [71]; this motivates the title of our software.

Multimodal image registration studies. Custom Python scripts were used to propagate alignment across imaging modalities. Manual landmark correspondences were used in all pairwise registrations, including full tissue alignments and within IMC-derived ROIs for the DFU, as well as for the PC TMA dataset. Using landmarks during the registration processes imparts bias to the alignment, as compared to fully unsupervised registration, by ensuring images more closely adhere to manually set correspondences across modalities. However, landmark overlap is a penalty in the optimization, rather than a strict constraint. Consequently, they guide the registration rather than forcing it to match potentially imperfect human input. All registrations involving MSI or IMC data were conducted using KNN $\alpha$-MI, with $\alpha$ = 0.99 and 15 nearest neighbors (S4 Note, **α-entropy**). All registrations aligning low-channel slides (toluidine blue IMC reference and H&E) were conducted using histogram-based MI after grayscale conversion for rapid processing. All data that underwent image registration were exported and stored as 32-bit NIfTI-1 images. IMC data were not transformed and were kept in 16-bit OME-TIF (F) format.

As data acquisition removed tissue context at regions of interest on the IMC full-tissue reference image, we first aligned full tissue sections, then used the coordinates of IMC regions to extract data from all modalities for fine-tuning. Manual landmarks were used to guide both registration steps. We accounted for alignment errors around IMC regions following full-tissue registration by padding regions prior to cropping. For full-tissue images in the DFU dataset, a two-step registration process was implemented by aligning images first using an affine model, and then with a nonlinear model parametrized by B-splines. Hierarchical Gaussian smoothing pyramids were used to account for resolution differences between image modalities, and stochastic gradient descent with random coordinate sampling was used for optimization. We additionally optimized final control point grid spacings for B-spline models (MSI to H&E alignment) and the number of hierarchical levels to include in pyramidal smoothing (MSI and IMC to H&E) (S1, S2, and S3 Figs). A final control point spacing of 300 pixels for nonlinear B-spline registrations of MSI data to corresponding H&E was found to balance correct alignment with unrealistic warping, which we identified visually and by inspecting the determinants of the spatial Jacobian matrices for values that deviated substantially from 1. H&E and IMC reference tissue registrations utilized a final grid spacing of 5 pixels.

For PC TMA cores, alignment was performed similarly, with grid spacing optimized for minimal warping. Manual landmark correspondences of at least 4 set points were used to guide the registration of each modality for each core.

Benchmark study to manual alignment. MIAAIM benchmarking was performed by comparing the improvement in $\alpha$-MI between the IMC and MSI modalities for each prostate cancer TMA core relative to manual landmark alignment using Elastix (S9 Fig). The same set of landmarks were used in both conditions (MIAAIM versus manual landmarking alone). Manual landmark registration was performed using a multiresolution, local-to-global alignment workflow in the same fashion as MIAAIM registration, where, for each TMA core, the IMC data was registered to the corresponding H&E data and then propagated to the IMC coordinate system by composition. The optimization and alignment were performed by first aligning modalities using a rigid transformation, followed by an affine transformation, and a final B-spline alignment. The $\alpha$-MI was quantified on the registered IMC and MSI modalities using the KNN graph-based estimator with $\alpha$ = 0.99 and 15 nearest neighbors on a 10x10 uniformly spaced grid. Pixels were only sampled from the nonzero intersection of TMA core masks after alignment with both approaches to reduce edge artifacts. The nearest-neighbor graph used to compute the $\alpha$-MI was built after principal component analysis (PCA) dimensionality reduction on each separate modality. The embedding dimensionality of each was determined by keeping 99% of the variation in the respective dataset.

**Microenvironmental correlation network analysis.** To calculate associations across MSI and IMC modalities, we used Spearman's correlation coefficient in the Python Scipy library. *m/z* peaks from MSI data with no correlations to IMC data with Bonferroni corrected P-values above 0.001 were removed from the analysis. Correlation modules were formed with hierarchical Louvain community detection using the Scikit-network package. The resolution parameter used

for community detection was chosen based on the elbow point of a graph plotting resolution vs. modularity of community detection results. UMAP's simplicial set, created with 5 nearest neighbors and the Euclidean metric, was used as input for community detection after inverse cosine transformation of Spearman's correlation coefficients to form metric distances [72]. Line plots to visualize the average trends of MSI correlation modules to IMC parameters were computed using exponential-weighted moving averages in the Pandas library in Python after standard scaling of IMC and MSI single-cell data. MSI moving averages were additionally min-max scaled to a range of 0–1 for plotting purposes. Differential correlations of variables *u* from MSI data and *v* from IMC data between conditions *a* and *b* were weighted by the maximal absolute correlation coefficient among both conditions and then ranked, following the method of Hsu et al [73]. Statistical significance levels of differential correlations were calculated using one-sided, Bonferroni corrected z-statistics after Fisher transformation.

**Single-cell segmentation (MIAAIM Probabilities and Segmentation modules).** To quantify parameters of single cells in IMC and registered MSI data within the DFU dataset, we performed cell segmentation on IMC ROIs using the pixel classification module in Ilastik (version 1.3.2) [30], which utilizes a random forest classifier for semantic segmentation. Ilastik is included in the MIAAIM software along with CellProfiler [31]. In addition, the following workflow is included outside of the noise-removal steps.

For each ROI, two 250 $\mu$m by 250 $\mu$m areas were cropped from IMC data and exported in the HDF5 format for use in supervised training. To ensure cropped areas were representative training samples, a global foreground threshold was calculated using Otsu thresholding on the Iridium (nuclear) stain with the Scikit-image Python library separately for each such region. Cropped regions were required to contain over 30% of pixels above their respective threshold.

Training regions were annotated for "background", "membrane", "nuclei", and "noise". Random forest classification incorporated Gaussian smoothing features, edges features (including Laplacian of Gaussian features, Gaussian gradient magnitude features, and difference of Gaussian features), and texture features (including structure tensor eigenvalues, and Hessian of Gaussian eigenvalues). The trained classifier was used to predict each pixels' probability of assignment to the four classes in the full images, and predictions were exported as 16-bit TIFF stacks.

To remove artifacts in cell staining, noise prediction channels were Gaussian blurred with a sigma of 2 and then subjected to Otsu thresholding with a correction factor of 1.3, which created a binary mask separating foreground (high pixel probability to be noise) from background (low pixel probability to be noise). The noise mask was used to assign zero values to the other three Ilastik probability channels (nuclei, membrane, background) to all pixels that were considered foreground in the noise channel. Noise-removed, three-channel probability images of nuclei, membrane, and background were used for single-cell segmentation in CellProfiler.

**Single-cell parameter quantification.** Single-cell parameter quantification for IMC and MSI data were performed using an in-house modification of the quantification (MCQuant) module in the multiple-choice microscopy software (MCMICRO) [22] to accept NIfTI-1 files after cell segmentation. This modification is the default in MIAAIM. IMC single-cell measures were transformed using 99th percentile normalization prior to downstream analysis. MSI data for the prostate cancer TMA were similarly transformed prior to analysis.

**Imaging mass cytometry cluster analysis.** Cluster analysis was performed in Python using the Leiden community detection algorithm with the leidenalg Python package. UMAP's simplicial set (weighted, undirected graph) created with 15 nearest neighbors and Euclidean metric was used as input to community detection.

**Single-cell spatial features.** Using each segmented cell's centroid as its spatial coordinates and Euclidean distances between cells as weights, a spatial nearest-neighbor graph was generated using Squidpy [74] by connecting each cell to its neighbors within a radius of 75 $\mu$m. A Gaussian kernel over the edges of this graph was then used to normalize the weights of edges starting from each cell to the standard deviation of the distances between that cell and all its connected neighbors. To produce single-cell neighborhood scores, i.e., spatial features, normalized weights were then summed across connected cells separately for each feature in the segmented cell-by-feature matrix. This can be viewed as a

spatial smoothing operator over the feature matrix for each core. These features were built for both the IMC and MSI parameters for each cell in each core.

**Logistic regression and feature selection of integrated multimodal data.** Single cell MSI/IMC profiles were normalized to a mean of 0 and a standard deviation of 1 prior to analysis. Multi-class logistic regression using a one-vs-rest approach (i.e., one model trained per class) was performed with elastic net regularization using the scikit-learn package. Regularization parameters were tuned using a grid search approach. To ensure a generalizable model and reduce overfitting due to unique signatures per TMA core, we performed three-fold cross validation on the set of TMA cores instead of the set of single cells. Model performance was evaluated using class balanced accuracy. Since the input data are normalized, model coefficients for each predictor represent the contribution of that predictor on the model's final output. This allows us to use the magnitude of model coefficients as a measure of feature importance and thereby rank the predictors. With this ranking, we trained and tested new models on subsets of ranked predictors from the single most important predictors to all predictors in increments of 50. To further refine the list of parameters, we used leave-one-out analysis to generate a set of generalized features based on accuracy degradation from the full model.

## Supporting information

**S1 Fig. Performance of dimensionality reduction algorithms for summarizing diabetic foot ulcer mass spectrometry imaging data. a.** Three mass spectrometry peaks highlighting tissue morphology were manually chosen (top) and were used to create and RGB image representation of the MSI data, which was converted to a grayscale image. The MSI grayscale image was then registered to its corresponding grayscale converted hematoxylin and eosin (H&E) stained section. The deformation field (middle), indicated by the determinant of its spatial Jacobian matrix, was saved to use downstream as a control registration. Three-dimensional Euclidean embeddings of the MSI data were then created using random initializations of each dimension reduction algorithm (bottom). These embeddings were then used to create an RGB image following the procedure above. The spatial transformation created by registering the manually identified peaks with the H&E image was then applied to dimension reduced grayscale images, aligning each to the grayscale H&E image. **b.** The mutual information between each aligned grayscale embedded image (n = 5 per method) and the grayscale H&E image was calculated using Parzen window histogram density estimation with a histogram bin width of 64. Plot is oriented so that results are consistent with the notion of a "cost function" in optimization contexts, where the goal is to minimize cost. Thus, larger negative values depict higher mutual information. UMAP consistently captures multi-modal information content with respect to the H&E data. **c.** Optimization of image registration between the grayscale version of manually identified mass spectrometry peaks and the grayscale H&E image (**a**, top) using mutual information as a cost function with external validation using dice scores on 7 manually annotated regions. Registration parameters used for the final registration used in panel **a** are indicated with dashed lines. Registration was performed by first aligning images with a multi-resolution affine registration (left). The transformed grayscale version of manually identified mass spectrometry peaks was then registered to the grayscale H&E image using a nonlinear, multi-resolution registration. **d.** Average neighborhood entropy (n = 5) of each pixel calculated within a 10-pixel disc across dimension reduction algorithms. Results show UMAP's ability to highlight structure in the tissue section. **e.** Manual annotations of grayscale H&E image used for validating registration quality with controlled deformation field in panel **a** used for panel **b** mutual information calculations. **f.** Cropped regions using the same spatial coordinates as **e** of manually annotated regions used to calculate the dice scores in **c**. Results show good spatial overlap across disparate annotations. **g.** Radar plots showing performance comparison of dimension reduction algorithms spanning a range of data representation – linear, nonlinear, local, and global data structure preservation (t-SNE, UMAP, PHATE, Isomap, NMF, PCA). Shown are mean values (n = 5) of algorithm runtime (top, log transformed), estimated manifold embedding dimensionality (right), noise robustness (bottom), and multi-modal mutual information to DFU MSI data (left). All plots are oriented so that larger values depict better algorithmic performance. Results show UMAP's ability to efficiently

capture data complexity with few degrees of freedom while balancing noise robustness with multi-modal information content contained in histology images. **h.** Intrinsic dimensionality of MSI data estimated by each dimension reduction method. Embedding errors (y-axes) are not comparable across plots. Plotted are the mean and standard deviation (n = 5) embedding errors across embedding dimensions 1–10. Convergence on y-axes indicates that increasing the dimension of the resulting embeddings no longer improves an algorithm's ability to capture data complexity. Results show that the intrinsic dimensionality estimated by nonlinear methods (t-SNE, UMAP, PHATE, Isomap) is far less than that of linear methods (NMF, PCA), meaning that fewer dimensions are needed to accurately describe the data set. **i.** Denoised manifold preservation (DeMaP) metric between Euclidean distances in resulting embeddings corresponding to non-peak-picked data and geodesic distances in ambient space (not dimension reduced after peak-picking) of corresponding peak-picked data. Results showing the mean and standard deviation DeMaP metric (Spearman's rho correlation coefficient) for all tested dimension reduction methods (n = 5). Nonlinear methods Isomap, PHATE, and UMAP all consistently preserve manifold structure without prior filtering of the data with consistent correlations greater than 0.85 across dimensions 2–10. **j.** Computational runtime for each algorithm across embedding dimensions 1–10. Plotted are the mean and standard deviation (n = 5) across each number of dimensions for each method. Nonlinear methods t-SNE and Isomap require longer run times than the nonlinear methods PHATE and UMAP. Linear methods require the least amount of run time; however, they fail to capture data complexity succinctly.
(TIF)

**S2 Fig. Performance of dimensionality reduction algorithms for summarizing prostate cancer mass spectrometry imaging data. a.** Same as in S1a Fig for prostate cancer tissue biopsy. b. Same as S1b Fig for prostate cancer tissue biopsy. c. Optimization of image registration between the grayscale version of manually identified mass spectrometry peaks and the grayscale H&E image (a, top) using mutual information as a cost function. Registration parameters used for the final registration used in a are indicated with dashed lines. Registration was performed by first aligning images with a multi-resolution affine registration (left). The transformed grayscale version of manually identified mass spectrometry peaks was then registered to the grayscale H&E image using a nonlinear, multi-resolution registration. d. Same as S1d Fig for prostate cancer tissue biopsy. e. Same as S1g Fig for prostate cancer tissue biopsy. f. Same as S1h Fig for prostate cancer tissue biopsy. g. Same as S1i Fig for prostate cancer tissue biopsy. Nonlinear methods Isomap, PHATE, and UMAP all consistently preserve manifold structure without prior filtering of the data with consistent correlations greater than 0.75 across dimensions 2–10. h. Same as S1j Fig for prostate cancer tissue biopsy.
(TIF)

**S3 Fig. Performance of dimensionality reduction algorithms for summarizing tonsil mass spectrometry imaging data. a.** Same as **S1 panel a** for tonsil tissue biopsy. **b.** Same as **S1 panel b** for tonsil tissue biopsy. Isomap and NMF consistently capture multi-modal information content with respect to the H&E data. **c.** Same as S2c Fig for tonsil tissue biopsy. d. Same as S1d Fig for tonsil tissue biopsy. e. Same as S1g Fig for tonsil tissue biopsy. f. Same as S1h Fig for tonsil tissue biopsy. g. Same as S1l Fig for tonsil tissue biopsy. h. Same as S1j Fig for tonsil tissue biopsy.
(TIF)

**S4 Fig. Spectral centroid landmarks recapitulate estimated optimal manifold embedding dimensionalities across tissue types and imaging technologies. a.** Sum of squared errors of exponential regressions fit to estimated optimal embedding dimensionality selections from spectral landmarks compared to full mass spectrometry imaging data sets across DFU, Prostate, and Tonsil tissues. Discrepancies between exponential regressions fit to the cross-entropy of landmark centroid embeddings and full data set embeddings approach zero as the number of landmarks increases. Dashed lines show MIAAIM's default selection of 3,000 landmarks for computing manifolds embedding dimensionalities. **b**. Same as **a** for subsampled pixels in imaging mass cytometry regions of interest.
(TIF)

**S5 Fig. UMAP embeddings of spatially subsampled imaging mass cytometry data with out-of-sample projection recapitulate full data embeddings while decreasing runtime in DFU samples. a.** Three-dimensional UMAP embedding (HDIprep compression) runtime plotted with respect to subsampling percentages of IMC ROIs from the DFU tissue biopsy (top). Procrustes transformation sum of squared errors after transforming subsampled embedding to the full pixel embedding across subsampling percentages (bottom). **b.** Comparison of RGB (red, green, blue) images created by reconstructing images from pixel embeddings on all data (top) versus subsampled data with subsequent out-of-sample projection and Procrustes transformation to align subsampled embedding to the full pixel embedding (bottom) (scale bars=80 $\mu$m). Subsampling percentages of images shown in bottom row of panel **b** correspond to MIAAIM default parameters that based on number of pixels in images.
(TIF)

**S6 Fig. UMAP embeddings of spatially subsampled imaging mass cytometry data with out-of-sample projection recapitulate full data embeddings while decreasing runtime in prostate cancer samples. a-b.** Same as S5 Fig for prostate tumor tissue biopsy IMC ROIs.
(TIF)

**S7 Fig. UMAP embeddings of spatially subsampled imaging mass cytometry data with out-of-sample projection recapitulate full data embeddings while decreasing runtime in tonsil samples. a-b.** Same as S5 Fig for tonsil tissue biopsy IMC ROIs.
(TIF)

**S8 Fig. MIAAIM image compression scales to large fields of view and high-resolution multiplexed image datasets by incorporating parametric UMAP. a.** Multiplex CyCIF image of lung adenocarcinoma metastasis to the lymph node (n=~100 million pixels, 0.65 $\mu$m/pixel resolution, 44 channels, 27 antibodies) and corresponding UMAP embedding and spatial reconstruction (shown are three UMAP channels of 4 channel estimated optimal embedding). Parametric UMAP compresses millions of pixels and preserves tissue structure across multiple length scales. **b.** Same as **S8a Fig** for tonsil CyCIF data (n=~256 million pixels, 0.65 $\mu$m/pixel resolution).
(TIF)

**S9 Fig. Benchmark of MIAAIM registration to conventional manual landmarking. a.** Relative improvement or reduction in $\alpha$-MI between IMC and MSI modalities for prostate cancer TMA cores registered with MIAAIM versus a manual landmark registration performed using Elastix. **b.** Visualizations of improvement in alignment using MIAAIM (top) compared to manual landmark alignment (bottom) for two chosen TMA cores (scale bars=150 $\mu$m, 25 $\mu$m). Representative UMAP pixel embedding channels and the H&E modality are shown to resolve fine-scale structure. Arrows (insets) point to subtle misalignment resulting from manual landmark registration that is corrected with MIAAIM.
(TIF)

**S10 Fig. Classification of prostate cancer TMA cores confusion matrix.** Train and test data were split at the TMA level, and logistic regression was used on single cell and neighborhood interaction measures of IMC/MSI signatures to predict Gleason score. A confusion matrix on classification performance was generated on the test data, showing that classification between benign and cancerous tissue was an easier task compared to classifying between tumor classes.
(TIF)

**S1 Table. Current imaging technologies anticipated to be compatible and those tested for compatibility with MIAAIM image compression and pre-processing for subsequent alignment.**
(DOCX)

**S2 Table. Comparison of MIAAIM to existing image registration approaches and software References.**
(DOCX)

**S3 Table. Prostate cancer TMA logistic regression model accuracy by modality.**
(DOCX)

**S1 Note. Combining MIAAIM with existing bioimaging analysis software.**
(DOCX)

**S2 Note. Notes on the HDIreg workflow's expected performance.**
(DOCX)

**S3 Note. HDIprep dimension reduction validation.**
(DOCX)

**S4 Note. Estimating pixel embedding dimensionality.**
(DOCX)

## Acknowledgments

We acknowledge PDTx for providing instrumentation access to a Standard Biotools Hyperion imaging mass cytometer. We thank Ruth Montgomery for input and facilitating access to a Standard Biotools Hyperion Imaging system through the Yale CyTOF core. We thank Benjamin J. Garcia for critical reading of the manuscript. We thank Enterprise Research Infrastructure & Services at Mass General Brigham for their in-depth support and provision of computing resources. Co-authors dedicate this publication in memory of Dr. John Iskra, a valued colleague, mentor, and friend.

## Author contributions

**Conceptualization:** Joshua M. Hess, Iulian Ilieş, Denis Schapiro, John J. Iskra, Mark C. Poznansky, Ruxandra F Sîrbulescu, Patrick M. Reeves.

**Data curation:** Joshua M. Hess, Richard K. Dzeng, Divya Mirgh, John Nam, Erin H. Seeley, Chin Lee Wu, Ruxandra F Sîrbulescu, Patrick M. Reeves.

**Formal analysis:** Joshua M. Hess, Richard K. Dzeng, Erin H. Seeley, Ruxandra F Sîrbulescu, Patrick M. Reeves.

**Funding acquisition:** Mark C. Poznansky, Ruxandra F Sîrbulescu, Patrick M. Reeves.

**Investigation:** Joshua M. Hess, Richard K. Dzeng, Divya Mirgh, John Nam, Erin H. Seeley, David E. Verrill, Michael S. Regan, Georgios Theocharidis, Chin Lee Wu, Aristidis Veves, Nathalie Y.R. Agar, Mark C. Poznansky, Ruxandra F Sîrbulescu, Patrick M. Reeves.

**Methodology:** Joshua M. Hess, Richard K. Dzeng, Iulian Ilieş, Denis Schapiro, Walid M. Abdelmoula, Georgios Theocharidis, Aristidis Veves, Nathalie Y.R. Agar, Ruxandra F Sîrbulescu, Patrick M. Reeves.

**Project administration:** Ann E. Sluder, Ruxandra F Sîrbulescu.

**Resources:** Joshua M. Hess, Divya Mirgh, Ruxandra F Sîrbulescu, Patrick M. Reeves.

**Software:** Joshua M. Hess, Richard K. Dzeng.

**Supervision:** Mark C. Poznansky, Ruxandra F Sîrbulescu, Patrick M. Reeves.

**Validation:** Richard K. Dzeng.

**Visualization:** Joshua M. Hess, Richard K. Dzeng, Mark C. Poznansky, Ruxandra F Sîrbulescu, Patrick M. Reeves.

**Writing – original draft:** Joshua M. Hess, Richard K. Dzeng, Iulian Ilieş, Mark C. Poznansky, Ruxandra F Sîrbulescu, Patrick M. Reeves.

**Writing – review & editing:** Joshua M. Hess, Richard K. Dzeng, Ann E. Sluder, Mark C. Poznansky, Ruxandra F Sîrbulescu, Patrick M. Reeves.

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
