## [Decision Letter · Decision Letter 0]

1 Dec 2025

PCOMPBIOL-D-25-01551

MIAAIM: Multi-Omics Image Integration with Dimensional Reduction for Tissue State Mapping

PLOS Computational Biology

Dear Dr. Reeves,

Thank you for submitting your manuscript to PLOS Computational Biology and apologies for the delay. After careful consideration, we feel that your manuscript has significant merit but does not fully meet PLOS Computational Biology's publication criteria as it currently stands. Therefore, we invite you to submit a revised version of the manuscript that addresses the points raised during the review process.  In particular, please address the concerns of the reviewers concerning quantitative benchmarking of your algorithm and provide better support for your use of a multi-modal analysis on experimental data.

Please submit your revised manuscript within 60 days Jan 31 2026 11:59PM. If you will need more time than this to complete your revisions, please reply to this message or contact the journal office at ploscompbiol@plos.org. Please include the following items when submitting your revised manuscript:

We look forward to receiving your revised manuscript.

Kind regards,

Joshua N. Milstein

Academic Editor

PLOS Computational Biology

Shayn Peirce-Cottler

Section Editor

PLOS Computational Biology

**Additional Editor Comments (if provided):**

**Journal Requirements:**

**Reviewers' comments:**

Reviewer's Responses to Questions

**Comments to the Authors:**

Reviewer #1: The classification of prostate cancer Gleason scores is a powerful application of the MIAAIM framework. The manuscript already contains the critical data showing that the integrated multimodal model outperforms models based on single modalities. This result is a cornerstone of the paper's argument, as it provides direct evidence that multimodal integration yields a more complete and predictive representation of tissue states.

To make this method more impactful, it is suggested add an experiment to the presentation of these results: consider adding a summary table or a simple bar chart in the main text that directly compares the final classification accuracy (e.g., class-balanced accuracy) for models trained on:

1. IMC features only (including single-cell and neighborhood interaction scores).

2. MSI features only (including single-cell and neighborhood interaction scores).

3. The fully integrated IMC + MSI model.

While this information is mentioned in the text, a dedicated figure or table would provide a clear visualization of the performance gain achieved through multimodal integration.

Reviewer #2: Summary:

This paper describes MIAAIM, an open-source and modular computational framework for aligning and integrating multimodal, high-dimensional tissue imaging. By combining concepts such as UMAP and entropic-graph-based mutual information, MIAAIM can align data from disparate modalities. The authors focus on aligning data from imaging mass cytometry (IMC), mass spectrometry imaging (MSI), and histology. They apply the framework to a large diabetic foot ulcer (DFU) biopsy and to multi-core prostate cancer tissue microarrays (TMAs), demonstrating MIAAIM’s ability to extract biologically meaningful multimodal single-cell and spatial features.

Major Comments:

My main concern is the lack of quantitative benchmarking. The α–mutual information alignment method is conceptually elegant, but registration quality is reported primarily through qualitative visual inspection, which is a serious limitation. Commonly used quantitative metrics such as Dice coefficient, target registration error (TRE), or mutual information gain relative to baseline methods (e.g., Elastix) should be reported. I cannot foresee a path toward publication without explicit comparisons. These results are critical to clarify MIAAIM’s robustness and, more importantly, justify its novelty relative to existing registration tools. Lastly, a brief discussion of false discovery rates or validation strategies would strengthen the biological claims.

Minor Comments:

• Standardize reference formatting (several in-text citations have unmatched parentheses).

• Ensure all acronyms are defined at first mention.

• Clarify how manual landmarks were used in DFU alignment and whether this step introduces bias.

**Have the authors made all data and (if applicable) computational code underlying the findings in their manuscript fully available?**

Reviewer #1: Yes

Reviewer #2: Yes

PLOS authors have the option to publish the peer review history of their article (what does this mean?). If published, this will include your full peer review and any attached files.

Reviewer #1: No

Reviewer #2: No

**Figure resubmission:**
---

## [Decision Letter · Decision Letter 1]

27 Apr 2026

Dear Dr. Reeves,

We are pleased to inform you that your manuscript 'MIAAIM: Multi-Omics Image Integration with Dimensional Reduction for Tissue State Mapping' has been provisionally accepted for publication in PLOS Computational Biology.

Best regards,

Joshua N. Milstein

Academic Editor

PLOS Computational Biology

Virginia Pitzer

Editor-in-Chief

PLOS Computational Biology

Reviewer's Responses to Questions

**Comments to the Authors:**

Reviewer #2: I am satisfied with the changes to the manuscript and believe the paper is ready for publications. Congratulations to the author for the great work.

**Have the authors made all data and (if applicable) computational code underlying the findings in their manuscript fully available?**

Reviewer #2: Yes

PLOS authors have the option to publish the peer review history of their article (what does this mean?). If published, this will include your full peer review and any attached files.

Reviewer #2: No

---

## [Editor Report · Acceptance letter]

PCOMPBIOL-D-25-01551R1

MIAAIM: Multi-Omics Image Integration with Dimensional Reduction for Tissue State Mapping

Dear Dr Reeves,

I am pleased to inform you that your manuscript has been formally accepted for publication in PLOS Computational Biology. Your manuscript is now with our production department and you will be notified of the publication date in due course.

With kind regards,

Anita Estes
